# Hypothesis Selection with Memory Constraints

**Maryam Aliakbarpour**
Department of Computer Science
Rice University
Houston, TX 77005
`maryama@rice.edu`

**Mark Bun**
Department of Computer Science
Boston University
Boston, MA 02215
`mbun@bu.edu`

**Adam Smith**
Department of Computer Science
Boston University
Boston, MA 02215
`ads22@bu.edu`

## Abstract

Hypothesis selection is a fundamental problem in learning theory and statistics. Given a dataset and a finite set of candidate distributions, the goal is to select a distribution that matches the data as well as possible. More specifically, suppose that we have sample access to an unknown distribution $P$ over a domain $\mathcal{X}$ that we know is well-approximated by one of a class of $n$ distributions (a.k.a. hypotheses), $\mathcal{H} := \{H_1, H_2, \ldots, H_n\}$. The goal is to design an algorithm that outputs a distribution $\hat{H} \in \mathcal{H}$ whose total variation distance from $P$ is nearly minimal. In this work, we study the hypothesis selection problem under memory constraints. We consider a model where samples from $P$ are presented in a stream and we access each sample $x$ via "PDF-comparison" queries that allow us to compare the probability densities of any pair of hypotheses at the domain point $x$ (i.e., is $H_i(x) < H_j(x)$?). This model allows us to study how much needs to be stored, at any point in time, about the portion of the stream seen so far. Our main result is an algorithm that achieves a nearly optimal tradeoff between memory usage and sample complexity. In particular, given $b$ bits of memory (for $b$ roughly between $\log n$ and $n$), our algorithm solves the hypothesis selection problem with $s$ samples, where $b \cdot s = O(n \log n)$. This result is optimal up to an $O(\log n)$ factor, for all $b$.

## 1 Introduction

Learning the probability density function of observed data is a fundamental question in statistics with numerous applications in machine learning. Variants of this problem have been studied for nearly a century. *Hypothesis selection* is a classic version of this problem where the goal is to learn a distribution within a pre-specified class. Let $\mathcal{H} := \{H_1, H_2, \ldots, H_n\}$ be a class of $n$ known distributions over the same domain $\mathcal{X}$. Suppose that we have sample access to an unknown distribution $P$ over $\mathcal{X}$ which is in, or a very close to, a member of $\mathcal{H}$. The goal is to design an algorithm that outputs a distribution $\hat{H} \in \mathcal{H}$ whose total variation distance to $P$ is close to that of the closest distribution in $\mathcal{H}$. A great deal of effort has been dedicated to solving this problem using a number of samples that does not depend on the domain size $|\mathcal{X}|$. Perhaps surprisingly, one can learn an unknown distribution $P \in \mathcal{H}$ with $O(\log n)$ samples, independent of the domain size. In contrast, learning an arbitrary distribution $P$ over $\mathcal{X}$ requires $\Omega(|\mathcal{X}|)$ samples. Refined guarantees for this problem have been studied extensively, building an understanding of the accuracy, sample

37th Conference on Neural Information Processing Systems (NeurIPS 2023).

complexity, and computational efficiency achievable, as well as the compatibility of hypothesis selection with robustness and privacy [Yat85, DL96, DL97, DL01, MS08, DDS12, DK14, DKS17, ADLS17, BKM19, CKM+19, BKSW19, GKK+20, ABH+20, BBK+21].

In this paper, we expand upon the emerging theory of learning with limited memory [SD15, Raz18, Sha14, DS18, DGKR19, SSV19, AMNW22, BBS22] by studying hypothesis selection from this new perspective. We strive to answer the following questions: Given a working memory of $b$ bits, how many data points (samples) do we need to solve the hypothesis selection problem? Prior work assumes that we can store all the samples in the memory, which can be quite large when we aim to learn a distribution over extremely large objects, such as, DNA sequences, gene expression data, text data, and brain image data.

We consider a model where samples are processed one at a time in a stream. Similar to models in [GR22, CFG+23, AF23], we access each sample $x$ via queries that allow us to compare the PDF of $H_i$'s at point $x$, and we measure the size of the memory retained between processing samples. Our main result is an algorithm that achieves a nearly optimal tradeoff between the memory size $b$ and the number of samples $s$. Given $b$ bits of memory (for $b$ roughly between $\log n$ and $n$), our algorithm solves the hypothesis selection problem with $s$ samples where $b \cdot s = O(n \log n)$. A result of Shamir [Sha14, Theorem 2] gives a class of $n$ hypotheses $\mathcal{H}$ such that any algorithm that learns unknown distributions in $\mathcal{H}$ requires $s \cdot b = \Omega(n)$. Our tradeoff is thus optimal up to a $O(\log n)$ factor.

## 1.1 Main result

Suppose we have a class of $n$ known distributions $\mathcal{H} := \{H_1, H_2, \ldots, H_n\}$ and an unknown distribution $P$. We aim to design a *proper learning* algorithm that outputs a distribution in $\mathcal{H}$ whose total variation distance to $P$ is comparable to that of the closest distribution in $\mathcal{H}$. The algorithm has access to a stream of i.i.d. samples drawn from $P$. At every given point, the algorithm can look at one sample, namely $x$, in the stream and query a *PDF-comparator* oracle by sending $i, j \in [n]$, and receiving a bit indicating if $H_i(x) < H_j(x)$. This is equivalent to asking whether a sample $x$ is in the *Scheffé set* of two distributions $H_i$ and $H_j$, which is defined as the set of elements to which $H_j$ assigns higher probability than $H_i$. The algorithm can also discard the current sample and move on to the next one. In addition to having access to the samples of $P$, we have more direct access to the known distributions in $\mathcal{H}$. The algorithm can query the probability masses of the Scheffé sets according to each $H_i$ as is customary in the literature [DL01, MS08]. This assumption can be relaxed; only estimates of such probabilities are needed, which can be obtained if we have sample access to $H_i$'s. Further details are available in Remark 3. To summarize our access model, our algorithm can make one of the following types of queries: 1) Request a new sample. 2) Query the PDF-comparator on the current sample $x$ and ask if $H_i(x) < H_j(x)$ for every pair of indices $i, j \in [n]$. 3) Asks for the probability mass of Scheffé set between $H_i$ and $H_j$ according to $H_i$ for every pair of indices $i, j \in [n]$.

The accuracy of our output distribution is measured with respect to how far the *best* distribution in $\mathcal{H}$ is from $P$. We define $\mathsf{OPT}(\mathcal{H}, P) := \min_{H \in \mathcal{H}} \|P - H\|_{\mathrm{TV}}$. When $\mathcal{H}$ and $P$ are clear from context, we denote this simply by $\mathsf{OPT}$. With this setup in mind, we define a *proper learning algorithm (with promise)* for hypothesis selection. The term "promise" refers to the extra parameter $\Gamma$ that the algorithm receives as input: the learner may assume that $\mathsf{OPT} \leq \Gamma$. (See Remark 2 for the case where $\Gamma$ is not provided.)

**Definition 1.1.** *Suppose $\mathcal{A}$ has sample access to an unknown distribution $P$. And, it can query the probabilities of the Scheffé sets according to each $H_i$ and a PDF-comparator for every pair of hypotheses in $\mathcal{H}$. Assume $\mathcal{A}$ is given an additional input parameter $\Gamma > 0$. We say algorithm $\mathcal{A}$ is an $(\alpha, \epsilon, \delta)$-proper learner with a promise for $\mathcal{H}$ if the following holds: for every distribution $P$, if $\Gamma \geq \mathsf{OPT}(\mathcal{H}, P)$ then, with probability at least $1 - \delta$ over its input sample (drawn i.i.d. from $P$) and internal coin tosses, $\mathcal{A}$ outputs a distribution $\hat{H} \in \mathcal{H}$ such that:*

$$\|P - \hat{H}\|_{TV} \leq \alpha \cdot \Gamma + \epsilon. \tag{1}$$

One can generally reduce $\epsilon$ and $\delta$ by increasing the number of samples taken or the time spent, whereas the value of $\mathsf{OPT}$, and hence $\Gamma$, is inherent to the problem instance. Therefore, it is more important and challenging to design algorithms that minimize the multiplicative parameter $\alpha$.

**Main theorem:** Our main result is a proper learner (with promise) that obtains a nearly optimal tradeoff between memory and data. Formally, we have the following theorem:

**Theorem 1.** *There exists a constant $c$ for which the following holds. Let $\mathcal{H}$ be an arbitrary class consisting of $n$ distributions, and let $\epsilon > 0$. For every natural number $b$ where $c \cdot (\log n + \log((\log n)/\epsilon)) \leq b \leq n$, there exists an $(\alpha = 9, \epsilon, \delta = 0.1)$-proper learner (with promise) for $\mathcal{H}$ using $b$ bits of memory and the following number of samples:*

$$
s = \begin{cases}
O\left(\dfrac{n \log n}{b} \cdot \dfrac{1}{\epsilon^2} \log\left(\dfrac{1}{\epsilon}\right)\right) & \text{when } b \leq \frac{n}{\log \log n}\,, \\[3mm]
O\left(\dfrac{n \log n}{b} \cdot \dfrac{1}{\epsilon^2} \log\left(\dfrac{\log n}{\epsilon}\right)\right) & \text{when } \frac{n}{\log \log n} \leq b \leq n\,.
\end{cases}
$$

Roughly speaking, this theorem says that given $b$ bits of memory that suffice to perform basic operations such as keeping track of indices and counting samples (i.e., $b = \Omega\left(\log n + \log((\log n)/\epsilon)\right)$), then $s \cdot b \approx O(n \log(n)/\epsilon^2)$ samples suffice to solve the hypothesis selection problem. A result of Shamir [Sha14, Theorem 2] gives a class of $n$ hypotheses $\mathcal{H}$ such that every algorithm that learns a random $P$ in $\mathcal{H}$ requires $s \cdot b = \Omega(n)$. Our tradeoff is thus optimal up to an $O(\log n)$ factor. We speculate that our result is tight even for the class considered in [Sha14], but we leave the proof of a tighter lower bound to future work.

**Remark 2.** *Both versions of the hypothesis selection problem (with and without a promise that $\mathsf{OPT} \leq \Gamma$) have been extensively studied (e.g. [DL01, DK14]). For simplicity, we present our result when $\Gamma$ is given a priori. Section C describes a reduction that yields a similar result when $\Gamma$ is not provided; the accuracy guarantee is analogous to that of Equation 1, but for slightly larger $\alpha$.*

## 1.2 Overview of our key ideas and other results

**Background on comparing two hypotheses:** Like many previous hypothesis selection algorithms, our algorithm relies on the ability to compare two hypotheses $H_1$ and $H_2$ based on the probabilities of their Scheffé set $S$. We estimate $P(S)$, and declare $H_i$ the winner for which $H_i(S)$ is closer to $P(S)$. This natural approach works especially well when $P \in \{H_1, H_2\}$ or when $P$ is much closer to one of the two hypotheses. Intuitively, a near-optimal hypothesis $H$ should win many comparisons against other hypotheses. A typical hypothesis selection algorithm will run a tournament to identify such an $H$. The primary advantage of using Scheffé based comparisons is that they are inexpensive to compute. Only $O(\log n/\epsilon^2)$ samples are sufficient to estimate the probability masses of all Scheffé sets within an error of $\epsilon$. This exceptional sample efficiency enables the hypothesis selection algorithms circumvent the lower bound of $\Omega(|\mathcal{X}|/\epsilon^2)$ for distribution learning in cases where we have prior knowledge that $P$ is close to a particular class of distributions.

That being said, the Scheffé based comparisons are not straightforward. One challenge is that these comparisons are not ideal: even if our estimates of the probabilities of the Scheffé sets were perfect, we might pick the hypothesis that is further from $P$ in total variation distance (because $S$ is just one event and might not ideally distinguish $P$ from either $H_1$ or $H_2$). Furthermore, comparisons are not transitive (again, because we measure only Scheffé set probabilities). Essentially the only useful property they have is this: if $H_1$ is $\Gamma$-close to $P$, it will win; if $H_1$ is much further from $P$ (by a constant factor) than both $\Gamma$ and $\|H_2 - P\|_{\mathrm{TV}}$, then it will lose. These issues make the outputs of comparisons hard to interpret and complicate their use for selection.

**Our terminology:** To elaborate on our approach, we start by defining a *acceptable* hypotheses, and our general terminology for referring to the quality of the hypothesis. We partition the hypotheses into three groups: 1) Excellent hypotheses, whose distance to $P$ is $\Gamma$. 2) Decent hypotheses, whose distance to $P$ is larger than $\Gamma$, but smaller than $3\Gamma + \epsilon$. These distances are set in a way that decent hypothesis may win against an excellent hypothesis. Decent hypotheses are challenging to deal with because they may fool us into believing that an excellent hypothesis is not a good output, yet later they may lose against a very far hypothesis. 3) Unacceptable hypotheses, whose distance to $P$ is larger than $3\Gamma + \epsilon$, which will lose to every excellent hypotheses. We say a hypothesis is *acceptable* if it is either excellent or decent. Generally, the goal is to find an acceptable hypothesis.

### 1.2.1 Random ladder tournament

The main technical tool we introduce is a simple algorithm we call the *random ladder tournament*, which is inspired by *ladder tournaments*, a type of tournaments used in games and sports. Our algorithm finds an acceptable hypothesis using a linear number of comparisons. The algorithm proceeds in rounds as follows. Holding on to a single hypothesis in each round, we sample a uniformly random hypothesis from $\mathcal{H}$ and compare it to the current hypothesis. The winner proceeds to the next round. Our novel analysis of this tournament shows that after $O(|\mathcal{H}|)$ rounds, either the final winner produced at the end of the tournament or a winner selected randomly along its trajectory will be acceptable with high probability.

**Proof overview:** As previously mentioned, comparisons are often far from ideal; Even when the excellent hypothesis wins at a certain step, there is no guarantee that it would remain the winner till the end. To evade this issue, we utilize our knowledge of the upper bound of $\mathsf{OPT}$, denoted by $\Gamma$, and modify the comparisons in a way that ensures the excellent hypothesis never loses once it wins. In our comparisons, we favor the current winner; If the distance of the current winner to $P$ on the Scheffé set is at most $\Gamma$, we declare the current winner as the winner (even when the opponent hypothesis seems closer). Clearly, the excellent hypothesis will never be more than $\Gamma$ away for $P$ on any set. Hence, it will never lose again.

However, what happens when the excellent hypothesis never wins? At each step, we pick each hypothesis in $\mathcal{H}$ uniformly at random, implying the excellent hypothesis is expected to be chosen in at least $O(1)$ steps in our algorithms. The fact that the random hypothesis loses to the current winners of those steps indicates that those winners must be acceptable hypotheses; otherwise, the excellent hypothesis would have won them. Additionally, the fact that the excellent hypothesis appears as the random hypothesis in $O(1)$ steps and loses to all of them, confirms that a constant fraction of the current winners are indeed acceptable. By combining these observations, we prove that the random ladder tournament will either find the excellent hypothesis at the end or encounter many acceptable hypotheses along the way.

**Implementing the tournament with memory constraints:** This algorithm is a highly effective technical tool for solving our problem. It can be implemented in a very memory-efficient manner. The algorithm processes the hypotheses in an "online" fashion, and it does not need to memorize the result of any past comparisons. At any given moment, it only remembers two hypotheses: the current hypothesis and the next randomly drawn hypothesis. And, if we draw fresh samples for every comparison, we can implement this algorithm by storing essentially $O(1)$ numbers but sampling $O(n\log(n)/\epsilon^2)$ data points. On the other extreme, we can implement this algorithm with very little data, i.e., $O(\log n/\epsilon^2)$ samples, but a large amount of memory by storing an appropriate "summary" of samples (e.g., the probabilities of all the Scheffé sets in $\tilde{O}(n^2)$ bits).

We leverage this flexibility in memory usage of the random ladder tournament and provide an algorithm that only uses $b$ bits of memory. We perform the comparisons in the random tournament in blocks of size $t$. For each block of comparisons, we draw new samples and store a summary of them. We pick the parameter $t$ in way that the summary fits in $b$ bits, and it is sufficient for us to perform the next the $t$ comparisons. For example, if $t \approx \sqrt{b}$, we can store the number of samples in all the Scheffé sets of $t$ hypotheses in $b$ bits. This summary is enough for us to perform any comparisons between those hypotheses. Combined with a novel summary—sorted lists, described in Section 2.2.2—we get a tradeoff of the form $s \cdot b \approx O(n\log^3(n)/\epsilon^4)$. In Section 3.2, we presented this tradeoff alongside another tradeoff (with better dependence to $\epsilon$, but worse dependence to $\log n$). Both of these tradeoff are worse than our main result, in a sense that $s \cdot b$ is larger by a factor of $O(\mathrm{poly}(\log(n), 1/\epsilon))$. However, the these results are yielding more accurate proper learners; For the random ladder tournament, $\alpha = 3$, while in our main result $\alpha = 9$.

**Simpler linear-time selection:** The random ladder tournament leads to new results for the hypothesis selection problem outside of the scope of our original motivation for this paper. If we ignore memory constraints and store every sample, the random ladder tournament algorithm is simultaneously near-optimal in terms of number of samples, time, and accuracy — to our knowledge, it is the first such algorithm. Specifically, it makes a linear number of comparisons and uses $O(\log n/\epsilon^2)$ samples from $P$, which are both optimal for worst-case choices of $\mathcal{H}$. Moreover, it finds a hypothesis with optimal accuracy under a promise. More precisely, it finds a hypothesis in $\mathcal{H}$ that is roughly

$3\,\Gamma$-close to $P$ given a parameter $\Gamma$ that is guaranteed to be at least $\mathsf{OPT}$. This is essentially optimal, as no proper learning algorithm can achieve accuracy better than $3\,\mathsf{OPT}$ [BKM19] even when $\Gamma = \mathsf{OPT}$ is given to the algorithm. Furthermore, with a simple adjustment to our algorithm, one can solve the hypothesis selection problem without any knowledge of $\Gamma$ in nearly linear time and with roughly the same number of samples and obtain a $5\,\mathsf{OPT}$-close hypothesis in $\mathcal{H}$. See Section C.

### 1.2.2 Improved tradeoff by hypothesis filtration

The flexibility of the random ladder tournament already results in memory-data tradeoffs, but these are suboptimal by a factor of $\mathrm{poly}(\log n, 1/\epsilon)$. To improve the result, we design an algorithm called *filter*, that picks an acceptable hypothesis with a modest chance. While this chance is not high, we can use the output of the filtration and run the random ladder tournament on these filtered hypotheses. Since the quality of the input hypotheses in this second round is better, the random ladder tournament can be implemented effectively on a smaller set $\mathcal{H}$ and results in a better tradeoff. Hence, we obtain our main result. See Section B.4 for the proof of our main results. Below, we give a short description of the filter algorithm. We defer the full description and the proofs to Section B.3.

**hypothesis filtration:** The best approach to finding an acceptable hypothesis drastically depends on the quality of the hypotheses in $\mathcal{H}$. For example, if there is no decent hypothesis in $\mathcal{H}$, any single elimination tournament will easily find an excellent hypothesis due to the fact that any comparison involving at least one excellent hypothesis will declare the excellent hypothesis the winner. On the other hand, if there are a lot of decent hypotheses, then there is a decent chance that a randomly selected hypothesis is decent, hence an acceptable one. The filtration algorithm works by combining these two observations. The algorithm randomly performs one of the following, each with probability $1/2$: 1) It draws a set of random hypotheses. Assuming there is no decent hypothesis in the set, it runs a *group elimination tournament* (see below), and outputs the result. 2) It outputs a random hypothesis in $\mathcal{H}$. One can show that if the decent hypotheses are scarce in $\mathcal{H}$, we have a decent chance of getting an excellent hypothesis and no decent hypothesis in the random set; hence, we find an excellent hypothesis via our group elimination algorithm. On the other hand, if the decent hypotheses are abundant, then we have a decent chance of picking one in the second step. By selecting each strategy with a probability $1/2$, we preserve half of the success probability in whichever case holds. While this probability is not a lot, it is larger than $1/n$ (the probability of picking the excellent hypothesis from $\mathcal{H}$). We take advantage of this fact, and show one can run the random ladder tournament in fewer rounds and obtain the desired result. See Section B.3 for more details.

**Group elimination tournament:** Our second technical tool is an algorithm that finds an excellent hypothesis if no decent hypothesis exists in $\mathcal{H}$. This algorithm is a combination of a single-elimination tournament and an *all-go-against-all* tournament. In this algorithm, at every step, we partition the hypotheses into multiple groups, run a previously known all-go-against-all tournament within each group (say minimum distance estimate [DL01, MS08]), and then send the winners of each group to the next round. If a group contains an excellent hypothesis, then in an all-go-against-all tournament, we will pick an excellent hypothesis as the winner. Hence, an excellent hypothesis will "bubble up" in each round and be declared the final winner. With a careful choice of group sizes and the number of rounds, this algorithm can be implemented using $O(n)$ bits of memory and $O(\log n)$ samples under the assumption that the hypotheses are indexed from 1 to $n$. A technical challenge that arises, however, is that we need to run this algorithm on a subset of roughly $O(b)$ random hypotheses (so that we can run it within a memory bound of $b$). However, storing $b$ random indices already requires $O(b \log n)$ space. We show that we can instead draw indices pseudorandomly using a pairwise independent generator that can be implemented in small space. See Section B.1 for more details.

## 1.3 Related work

Prior to our work, numerous studies have investigated the problem of *hypothesis selection* in various settings. In their seminal work, Yatracos [Yat85], and Devroye and Lugosi [DL96, DL97, DL01] present algorithms to find a close distribution in $\mathcal{H}$ based on only estimation of the probabilities of the Scheffé sets of pairs of hypothesis in $\mathcal{H}$. These approaches suggest that we only need accurate estimation of the probability that $P$ assigns to $O(|\mathcal{H}|^2) = O(n^2)$ sets, resulting in sample complexity proportional to $O(\log n)$ instead of $\Omega(|\mathcal{X}|)$. For a comprehensive overview, see Chapter 6 in [DL01].

Mahalanabis and Stefankovic in [MS08] provide an algorithm with linearly many probes to $P$ and $\alpha = 3$ but they require an exponential time pre-processing of the class $H$.[1] Daskalakis and Kamath

---

[1] It is worth noting that we do not have any pre-processing step in this paper.

in [DK14] give a nearly-linear time algorithm that given an upper bound on $\mathsf{OPT} \leq \Gamma$, returns $\hat{H}$ such that $\|\hat{H} - P\|_{\mathrm{TV}} \leq \alpha \cdot \Gamma + \epsilon$ for some large constant $\alpha = 512$. Acharya et al. in [AJOS14, AFJ$^+$18] provide a $O(n \log n)$ time algorithm that finds a hypothesis with accuracy guarantee of $\alpha = 9$. For proper learners, Bousquet et al. in [BKM19] showed that the best $\alpha$ in Equation 1 that one can hope for is $\alpha = 3$ as long as the sample complexity does not grow with the domain size $|\mathcal{X}|$. Aamand et al. in [AAC$^+$23] also study statistical computational tradeoffs (time-data) for the hypotheses over *discrete domains*. Unlike our work, their setting allows pre-processing of the class $\mathcal{H}$ in polynomial time.

There is a long history of research focusing on a special case of this problem where $\mathcal{H}$ is a *specific structured class of distributions* such as mixtures of Gaussians [KMV12, DK14, SOAJ14, KSS18, ABM18, ABH$^+$20], histograms [Pea95, CDSS14, CDKL22], and polynomials [ADLS17]. The abstract hypothesis selection we study here is commonly used as a subroutine in solving these problems (usually in conjunction with some sort of a cover method). For a survey of results, see [Dia16].

Lastly, there is another field of study that tackles a question similar to ours: the problem of sorting items with noisy comparisons. One can view hypothesis selection as the task of finding the minimum item. With the exception of [AFJ$^+$18] that we have discussed above, these probabilistic noise models do not capture the geometric structure presenting in our problem. Therefore, we did not find any of these result yielding an immediate solution to our problem.

## 2 Preliminaries

### 2.1 Notation

Let $[n]$ denote the set $\{1, 2, \ldots, n\}$. Suppose that we have a probability distribution $P$ over a domain $\mathcal{X}$. For element $x$ in $\mathcal{X}$, $P(x)$ denotes the probability of $x$. We assume $\mathcal{X}$ is the domain of all the probability distributions in this article. For any subset of the domain $S$, $P(S)$ denotes the sum of the probabilities of all the elements in $S$. We denote the total variation distance between two distributions $P_1$ and $P_2$ by $\|P_1 - P_2\|_{\mathrm{TV}}$, and it is defined as $\max_{S \subseteq \mathcal{X}} |P_1(S) - P_2(S)|$ where $S$ is any measurable subset of the domain. We say $P$ is $\epsilon$-*close* to $Q$ iff $\|P - Q\|_{\mathrm{TV}}$ is at most $\epsilon$. Also, we say $P$ is $\epsilon$-*far* from $Q$ iff $\|P - Q\|_{\mathrm{TV}}$ is greater than $\epsilon$. We denote the Scheffé set of two distributions as follows: $\mathcal{S}(H_1, H_2) := \{x \in \mathcal{X} \mid H_1(x) < H_2(x)\}$.

### 2.2 Basic tools

In this section, we focus on our primary tools we have used throughout this article. First, we start with a comparisons algorithm that allows us to compare two hypotheses. Next, we use the fact that one can estimate the probabilities of the $O(n^2)$ many Scheffé sets up to error $c \cdot \epsilon$ (for some small constant $c < 1$) with probability at least $1 - \delta$ using $O(\log(n/\delta)/\epsilon^2)$ samples from $P$.

#### 2.2.1 Comparing two hypotheses

Our algorithms works based on a basic operation that allows us to compare two hypotheses $H_1$ and $H_2$ based on the probabilities of their Scheffé sets. We pick the hypothesis that appears to be closer to $P$ and declare it the *winner* (and the other hypothesis the *loser*). The challenging part is that these comparisons are not ideal. As in, even when our estimations of the probabilities of the Scheffé sets are accurate enough, we may pick a hypothesis that is further. However, one can guarantee that if the distance of the further hypothesis among $H_1$ and $H_2$ is much worse than the distance of the closer one, the comparison procedure would never select it as the winner. The algorithm is given three parameters: 1) $\Gamma$: an upper bound for $\mathsf{OPT}$; 2) $\epsilon$: indicating the estimation error for the Scheffé set probabilities is $\Theta(\epsilon)$; and 3) the confidence parameter $\delta$. Upon invoking this algorithm, it determines the result of the comparison with the guarantees formalized in the following lemma:

**Lemma 2.1.** *Upon receiving three parameters: $\Gamma$, $\epsilon$, and $\delta$, Algorithm 1 uses $O(\log(1/\delta)/\epsilon^2)$ samples and satisfies the following guarantees with probability at least $1 - \delta$:*

- *If $H_1$ is $\Gamma$-close to $P$, then the algorithm returns $H_1$.*

- *If $H_1$ is $(\max(\Gamma, \|H_2 - P\|_{TV}) + 2\|H_2 - P\|_{TV} + \epsilon)$-far from $P$, then the algorithm returns $H_2$.*

The proof of this lemma is provided in Section D.

---

**Algorithm 1** Choosing between two hypotheses

---

1: **procedure** COMPARE($H_1$, $H_2$, $\Gamma$, $\epsilon$, $\delta$, sample access to $P$)
2:     Draw $m = \Theta(\log(1/\delta)/\epsilon^2)$ samples from $P$
3:     $\hat{q} \leftarrow$ fraction of samples from $P$ in $\mathcal{S}(H_1, H_2)$ using the PDF-comparator.
4:     $p_1 \leftarrow H_1\left(\mathcal{S}(H_1, H_2)\right)$.
5:     $p_2 \leftarrow H_2\left(\mathcal{S}(H_1, H_2)\right)$.
6:     $\epsilon' \leftarrow \epsilon/2$
7:     **if** $|p_1 - \hat{q}| \leq \Gamma + \epsilon'$ or $|p_1 - \hat{q}| \leq |p_1 - \hat{q}|$ **then**
8:         Output $H_1$.                                 $\triangleright$ $H_1$ is the winner.
9:     **else**
10:        Output $H_2$.                                $\triangleright$ $H_2$ is the winner.

---

**Remark 3.** *While we have assumed that we have access to the probabilities of the Scheffé sets in the above algorithm, this assumption is not necessary. In fact, one can obtain similar guarantees to Lemma 2.1 as long as we have estimates of these probabilities up to accuracy $c \cdot \epsilon$ for a sufficiently small constant $c < 1$. Hence, our result can be adjusted to the setting that we have sample access to $H_i$'s instead.*

**Our terminology based on comparisons:** We say a distribution $H$ is *excellent* iff $\|H - P\|_{\mathrm{TV}}$ is $\Gamma$. We usually use $H^*$ to denote an excellent hypothesis. We say a distribution $H$ is *decent* iff $\|H - P\|_{\mathrm{TV}}$ is greater than $\Gamma$, but smaller than $3\Gamma + \epsilon$. It implies that a decent hypothesis may win an excellent $H^* \in \mathcal{H}$. We say a distribution $H$ is *acceptable* iff it is excellent or decent. And, we say a distribution $H$ is *unacceptable* iff it is not acceptable. Note that it is impossible for an unacceptable hypothesis to win an excellent hypothesis with probability greater than $\delta$. Throughout this article, when we say *valid comparisons*, we refer to an event in which the Scheffé sets are estimated accurately, so the conditions in Lemma 2.1 hold. Since we have only $O(n^2)$ many Scheffé sets, using $O(\log(n/\delta)/\epsilon^2)$ samples is enough to assume all the comparisons are valid with a probability of at least $1 - \delta$.

### 2.2.2 Space efficient sample summaries for comparisons within a batch of hypotheses

In order to obtain sample efficiency, we need sufficient information about samples that enables us to reuse the same set of samples for multiple comparisons. Below, we describe two approaches to *summarize* the sample set.

**Scheffé counts.** Observe that to compare two hypotheses $H_1$ and $H_2$, we only use the count of the number of samples in their Scheffé set $\mathcal{S}(H_1, H_2)$. Hence, instead of storing samples directly, for every pair of hypotheses in $\mathcal{H}$, we store the number of samples in their Scheffé set. We refer to this number as the *Scheffé counts*. If we have $m$ samples, we can store all the $O(n^2)$ Scheffé counts using $O(n^2 \log m)$ bits.

**Storing each sample via a sorted list.** In this approach, we store a succinct summary of each sample that allows us to infer its membership in each Scheffé set. This information suffices to perform the comparisons we need later. To store a sample $x$, we save an ordered list of hypotheses that is sorted according to the probability of $x$ (the PDF at point $x$). That is, we store a list of indices $i_1, i_2, \ldots, i_n$ such that $H_{i_1}(x) \leq H_{i_2}(x) \leq \cdots \leq H_{i_n}(x)$ (use hypothesis indices to break the tie). The summary of a single sample can be computed in $O(n \log n)$ time and takes $O(n \log n)$ bits of space to store. Now, to compare the PDF of two hypotheses $H_i$ and $H_j$, we simply can find $i$ and $j$ in the list and check whether $i$ precedes $j$ in the list.[2] Alternatively if we want to compare PDF's faster, we can store the indices in another list call $d_x$ that maps indices to list positions. We set $d_x[i_\ell] = \ell$ for every $\ell \in [n]$. In this case, to compare the PDF's, we can simple check if $d_x[i] < d_x[j]$.

### 2.2.3 All-go-against-all tournament

There are standard techniques in the literature to solve hypothesis selection problem when the estimates of all the Scheffé sets are available. See Chapter 6 in [DL01] and [MS08]. These approaches

---

[2]It is worth noting that we may not answer the PDF-comparison correctly when $H_i(x) = H_j(x)$. However, since $x$ does not contribute to the discrepancy between $H_i$ and $H_j$, its inclusion (or exclusion) to the Scheffé set is inconsequential.

---

**Algorithm 2** Random ladder tournament

---

1: **procedure** RANDOM LADDER TOURNAMENT($\mathcal{H}$, $p_0$, $\Gamma$, $\epsilon$, sample access to $P$ and $\mathcal{D}$)
2:     $k \leftarrow \lceil 2000/p_0 \rceil$
3:     $Q \leftarrow$ an empty list.
4:     $W_0 \leftarrow \emptyset$                                              $\triangleright$ Assume $W_0$ is a hypothesis that always loses.
5:     **for** $i = 1, 2, \ldots, k$ **do**
6:         $R_i \leftarrow$ a random hypothesis drawn from $\mathcal{D}$
7:         $W_i \leftarrow$ COMPARE($W_{i-1}$, $R_i$, $\Gamma$, $\epsilon$, $1/(100\,k)$)
8:         Add $W_i$ to $Q$, with probability $p_0$.
9:     Add $W_k$ to $Q$.
10:     Output a random hypothesis in $Q$.

---

suggest that we only need accurate estimation of the probability that $P$ assigns to $O(|\mathcal{H}|^2) = O(n^2)$ sets, therefore, the sample complexity can be proportionate to $O(\log n)$ instead of $O(|\mathcal{X}|)$. For a formal statement, see Fact D.1.

## 3   Random ladder tournament

In this section, we focus on the *Random ladder tournament* which is a proper learner (with promise) of $P$ in a finite class of $n$ hypotheses, $\mathcal{H}$. The algorithm is given a parameter $\Gamma$ as an upper bound for $\mathsf{OPT}(\mathcal{H}, X)$. It outputs $\hat{H} \in \mathcal{H}$ such that, with high probability we have: $\|\hat{H} - P\|_{\mathrm{TV}} \leq 3\,\Gamma + \epsilon$. Bousquet et al. in [BKM19] have shown that no proper learner can achieve an error better than $3\,\mathsf{OPT} + \epsilon$ unless the sample complexity grows with the domain size $|\mathcal{X}|$. Their lower bound holds even when the algorithm is provided with $\Gamma = \mathsf{OPT}$. Hence, the factor $\alpha = 3$ above is optimal.

For clarity in our presentation, in Section 3.1, we present the random ladder tournament assuming that: 1) We set the confidence parameter to be a small constant ($\delta = 0.1$). 2) the algorithm can call the COMPARE subroutine without specifying how this subroutine is implemented. In Section A.1, we show one can modify this algorithm to work for arbitrary small confidence parameter $\delta$. Also, in Section 3.2, we discuss how one can implement this algorithm with memory constraints. We use the sample summary approaches described in Section 2.2.2, and we obtain two (sub-optimal) memory-data tradeoffs for this algorithm.

### 3.1   Random ladder tournament with no memory constraints

In this section, we present the random ladder tournament that solves the hypothesis selection problem while a parameter $\Gamma$ is given to the algorithm as an upper bound for $\mathsf{OPT}$. For this result, we assume there is a *meta distribution* over $\mathcal{H}$, denoted by $\mathcal{D}$, with a non-negligible probability $p_0$ to draw a hypothesis that is $\mathsf{OPT}$-close. The algorithm considers hypotheses one at a time where at step $i$, we draw $R_i$ from the meta distribution. While later we use this algorithm with other meta distributions, it may help the reader to view the meta distribution as a uniform distribution over a set of $n$ hypotheses, $\mathcal{H}$. In this case, if we draw a random hypothesis from $\mathcal{H}$, it is guaranteed to be $\Gamma$-close to $P$ with probability $p_0 \geq 1/n$. Our algorithm finds a sufficiently close hypothesis using $\Theta(1/p_0)$ comparisons.

At a high-level, our algorithm goes through a list of randomly drawn hypotheses from $\mathcal{D}$, compares them, and keeps track of the *current winner* hypothesis. We denote the current winner hypothesis at step $i$ by $W_i$. Initially, we start with $W_0$ being equal to a fake hypothesis that loses to any other hypothesis. Then at every step, we take a randomly drawn hypothesis $R_i$ and compare it with $W_{i-1}$. We set the current winner of step $i$, $W_i$, be the winner of a comparison between $W_{i-1}$ and $R_i$. We show that after $k = \Theta(1/p_0)$ steps either the final winner, $W_k$, is an $\Gamma$-close hypotheses, or many of $W_i$'s that we have encountered are close to $P$. That is, either $W_k$ is an excellent hypothesis or a random $W_i$ is an acceptable choice. To exploit this fact at every step, we add each $W_i$ to a list, namely $Q$, with some small probability. At the end, we also add $W_k$ to $Q$. We prove a random hypothesis in $Q$ will be close to $P$ with high probability. Algorithm 2 shows the formal description of our approach, and we prove its performance in Theorem 4. Figure 1 illustrates a visual representation of the algorithm.

**Theorem 4.** *Suppose we can draw i.i.d. random hypotheses from an arbitrary meta distribution $\mathcal{D}$ over $\mathcal{H}$. And, we have a hypothesis $P$ that we aim to learn properly in $\mathcal{H}$. Assume we are given parameters $p_0$ and $\Gamma$ such that the probability that a random hypothesis is $\Gamma$-close to $P$ is at least $p_0$. For any $\epsilon \in (0, 1)$, Algorithm 2 is $(\alpha = 3, \epsilon, \delta = 0.1)$-proper learner (with promise) for the class $\mathcal{H}$.*

*Proof.* First, note that each comparison, invoked in Line 7, will satisfies the properties of Lemma 2.1 with probability at least $1 - 1/(100 k)$. Hence, using Lemma 2.1 and the union bound, one can assume with probability 0.99 for all the comparisons we have:

- If $W_i$ is $\Gamma$-close to $P$, it will not lose; it remains as the current winner for the rest of the steps: $W_i = W_{i+1} = \cdots = W_k$.

- If $W_{i-1}$ is $(3\Gamma + \epsilon)$-far from $P$, and $R_i$ is $\Gamma$-close to $P$, then $W_i$ is equal to $R_i$.

For the rest of this proof, we fix the response of comparisons for which the above conditions hold. The randomness used in the rest of this proof only depends on the randomness in the meta distribution over hypotheses and the internal coin tosses of the algorithm.

Our goal is to show that most of the hypotheses in the list, $Q$, are acceptable, so a randomly selected hypothesis in $Q$ will be acceptable as well. Our first step is to show that the expected number of acceptable hypotheses in $Q$ is high compared to the size of $Q$. For each $i$ in $[k]$, we define an indicator random variable $I_i$ that is one if we add an acceptable hypothesis to the $Q$ at step $i$ in Line 8, and zero otherwise. Also, we define another indicator variable $I_{k+1}$ corresponding to the event that the final $W_k$, which we added to $Q$ in Line 9, is an acceptable hypothesis. More formally, we have:

$$I_i := \mathbb{1}_{W_i \text{ is acceptable, and we add } W_i \text{ to } Q \text{ in Line 8.}} \qquad \forall i \in [k],$$
$$I_{k+1} := \mathbb{1}_{W_k \text{ is acceptable.}} \cdot$$

Clearly, the sum of $I_i$'s indicates the number of acceptable hypotheses in $Q$, so we focus on finding a lower bound for the expected value of this quantity. Let $\alpha_i$ be the probability of $W_i$ being acceptable. Without loss of generality, we set $\alpha_0$ equal to zero. It is not hard to see that: $\mathbf{E}[I_i] = p_0 \cdot \alpha_i$ for $i \in [k]$. Moreover for the last indicator variable, we have:

$$\mathbf{E}[I_{k+1}] = \mathbf{Pr}[W_k \text{ being acceptable}] \geq \mathbf{Pr}[\|W_k - P\|_{\mathrm{TV}} \leq \Gamma]$$
$$= \sum_{i=1}^{k} \mathbf{Pr}[W_{i-1} \text{ loses to } R_i, \text{ and } \|R_i - P\|_{\mathrm{TV}} \leq \Gamma, \text{ and } R_i \text{ does not lose to } R_{i+1}, R_{i+2}, \ldots, R_k].$$

Assuming the pairwise comparisons are done perfectly, $R_i$ being $\Gamma$-close to $P$, automatically implies that $R_i$ does not lose to any of the hypotheses $R_{i+1}, \ldots, R_k$. Thus, we have:

$$\mathbf{E}[I_{k+1}] \geq \sum_{i=1}^{k} \mathbf{Pr}[W_{i-1} \text{ loses to } R_i, \text{ and } \|R_i - P\|_{\mathrm{TV}} \leq \Gamma]$$
$$= \sum_{i=1}^{k} \mathbf{Pr}[W_{i-1} \text{ loses to } R_i \mid \|R_i - P\|_{\mathrm{TV}} \leq \Gamma] \cdot \mathbf{Pr}[\|R_i - P\|_{\mathrm{TV}} \leq \Gamma].$$

Note that the probability of $\|R_i - P\|_{\mathrm{TV}} \leq \Gamma$ is at least $p_0$. Now given that $R_i$ is $\Gamma$-close to $P$, it will certainly win any hypotheses that is $(3\Gamma + \epsilon)$-far from $P$. Thus, the event that $W_i$ is $(3\Gamma + \epsilon)$-far must have a lower probability than the event that $W_{i-1}$ loses to $R_i$. Therefore, we obtain the following lower bound:

$$\mathbf{E}[I_{k+1}] \geq \sum_{i=1}^{k} \mathbf{Pr}[\|W_{i-1} - P\|_{\mathrm{TV}} > (3\Gamma + \epsilon) \mid \|R_i - P\|_{\mathrm{TV}} \leq \Gamma] \cdot p_0$$
$$= \sum_{i=1}^{k} \mathbf{Pr}[W_{i-1} \text{ being unacceptable} \mid \|R_i - P\|_{\mathrm{TV}} \leq \Gamma] \cdot p_0.$$

Recall that we pick $R_i$ independently from all the previous hypothesis, $R_1, R_2, \ldots, R_{i-1}$, so one can say $W_{i-1}$ is independent of $R_i$. This implies:

$$\mathbf{E}[I_{k+1}] \geq \sum_{i=1}^{k} \mathbf{Pr}[W_{i-1} \text{ being unacceptable}] \cdot p_0 = \sum_{i=1}^{k} (1 - \alpha_{i-1}) \cdot p_0.$$

Putting it all together, the expected value of the sum of $I_i$'s is:

$$\mathbf{E}\left[\sum_{i=1}^{k+1} I_i\right] \geq \sum_{i=1}^{k} p_0 \cdot \alpha_i + (1 - \alpha_{i-1}) \cdot p_0 = k \cdot p_0 + p_0 \cdot (\alpha_k - \alpha_0) \geq k \cdot p_0.$$

The last inequality above is due to the fact that we set $\alpha_0$ to zero. Note that the above equation states that the expected number of acceptable hypotheses in $Q$ is at least $k \cdot p_0$. On the other hand, the expected number of hypotheses in $Q$ is $k \cdot p_0 + 1$. Thus, the expected number of unacceptable hypotheses in $Q$ is at most one. Now, by Markov's inequality, the probability that we have more than 50 unacceptable hypotheses in $Q$ is at most 0.02. And, by the Chernoff bound, we know that the probability of having less than 1000 many hypothesis in $Q$ is at most:

$$\mathbf{Pr}[\#\text{hypotheses in } Q < 1000] = \mathbf{Pr}\left[\mathbf{Bin}(k, p_0) < \frac{k \cdot p_0}{2}\right] \leq \exp\left(-\frac{k \cdot p_0}{8}\right) \leq 10^{-100}.$$

It is not hard to see that the ratio of unacceptable hypotheses to the total number of hypotheses in $Q$ is at most 50/1000=0.05. Therefore, a random hypothesis in $Q$ is acceptable with probability at least 0.95. Hence, by the union bound the total error probability is bounded by:

$$\mathbf{Pr}[\text{Outputting an unacceptable hypothesis}] \leq \mathbf{Pr}[\text{Outputting an unacceptable hypothesis } | \text{ valid comparisons}]$$
$$+ \mathbf{Pr}[\text{at least one invalid comparisons}] \leq 0.05 + 0.01 < 0.1$$

Therefore, the proof is complete. □

## 3.2 Memory-data tradeoffs of random ladder tournament

As we have discussed, one of the advantages of Algorithm 2 is its flexibility in the usage of memory and data. In the following, we describe a tradeoff that we can obtain for this algorithm using the sample summaries presented in Section 2.2.2. At a high-level, the following is how we achieve the tradeoff: suppose we have $b$ bits of memory. We choose the largest integer $t$ so that we can store the sample summary needed to compare $t$ hypotheses. Then, we run the random ladder tournament while we draw new samples and refresh the sample summary at every $t$ step. Our two described sample summaries Scheff'e counts and the sorted list lead to the following memory-data tradeoffs. Although these tradeoffs are not as tight as our main result (by factors of $\log n$ and $1/\epsilon$), they have better accuracy guarantees ($\alpha = 3$ instead of $\alpha = 9$ in our main result).

**Lemma 3.1.** *Suppose we have $n$ hypotheses in $\mathcal{H}$. For every $p_0 \geq 1/n$, $\epsilon$, and an integer $t$ between two and $k = \Theta(1/p_0)$, one can implement Algorithm 2 in such away that it uses:*

$$s = O\left(\frac{1}{t \, p_0} \cdot \frac{\log p_0^{-1}}{\epsilon^2}\right) \text{ samples, and } \quad b = O\left(t^2 \log\left(\frac{\log p_0^{-1}}{\epsilon}\right) + t \log n\right) \text{ bits of memory.}$$

**Lemma 3.2.** *Suppose we have $n$ hypotheses in $\mathcal{H}$. For every $p_0 \geq 1/n$, $\epsilon$, and an integer $t$ between two and $k = \Theta(1/p_0)$, one can implement Algorithm 2 in such away that it uses:*

$$s = O\left(\frac{1}{t \, p_0} \cdot \frac{\log p_0^{-1}}{\epsilon^2}\right) \text{ samples, and } \quad b = O\left(t \cdot \frac{\log(p_0^{-1}) \cdot \log n}{\epsilon^2} + t \log n\right) \text{ bits of memory.}$$

For the proofs of the above lemmas, see Section A.2.

## Acknowledgement

MA was supported by NSF awards CNS-2120667, CNS-2120603, CCF-1934846, and BU's Hariri Institute for Computing. This work was predominantly done while MA was affiliated with Boston University and Northeastern University. MB was supported by NSF awards CCF-1947889 and CNS-2046425, and a Sloan Research Fellowship. AS was supported in part by NSF awards CCF-1763786 and CNS-2120667 as well as Faculty Awards from Google and Apple.

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

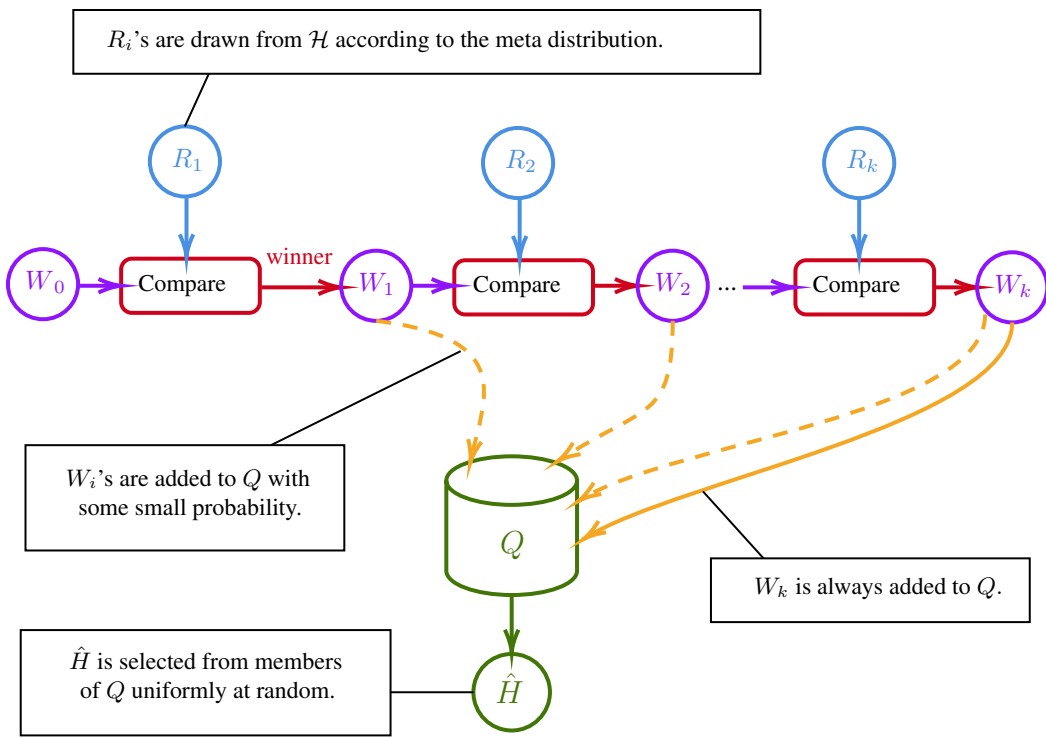

Figure 1: Random ladder tournament

## A  Random ladder tournament (cont.)

Figure 1 illustrates a visual representation of the algorithm.

### A.1  Amplifying the confidence parameter

In this section, we describe how one can amplify the confidence parameter of the random ladder tournament. Note that to keep $\alpha = 3$, we should avoid any two-step process that comes up with a collection of hypotheses to narrow down the possible choices and then pick an acceptable hypothesis among them. Such approaches generally lead to $\alpha = 9$ or higher. Here, we argue that if we run the random ladder tournament for a longer time, a random hypothesis in $Q$ will be acceptable with very high probability. In particular, if we set $k$, the number of steps equal to $\Theta(1/(\delta\, p_0))$ (instead of setting $k = \Theta(1/p_0)$), we find an acceptable hypothesis with probability at least $1 - \delta$.

**Corollary A.1.** *Suppose that we can draw i.i.d. random hypotheses from an arbitrary meta distribution $\mathcal{D}$ over $\mathcal{H}$. And, we have a hypothesis $P$ that we aim to learn properly in $\mathcal{H}$. Assume we are given parameters $p_0$ and $\Gamma$ such that the probability that a random hypothesis drawn from the meta distribution is $\Gamma$-close to $P$ is at least $p_0$. For any $\epsilon, \delta \in (0,1)$, Algorithm 2 can be modified to an $(\alpha = 3, \epsilon, \delta)$-proper learner (with promise) for class $\mathcal{H}$ by setting $k := \lceil 2\lceil 2/\delta\rceil / p_0\rceil = \Theta(1/(\delta\, p_0))$.*

*Proof.* Note that $\delta$ in Theorem 4 is 0.1. Hence, it suffices to focus on $\delta < 0.1$. As we have shown in the proof of Theorem 4, the expected number of unacceptable hypotheses in $Q$ is at most one. Let $\hat{H}$ be a random hypothesis in $Q$. Let $t := \lceil 2/\delta\rceil$, and set $k$ to $\lceil 2t/p_0\rceil$. It is not hard to see that the probability of $\hat{H}$ being unacceptable is bounded by $\delta$:

$$\mathbf{Pr}\Big[\hat{H} \text{ is } (3\,\Gamma + \epsilon)\text{-far}\Big] = \mathbf{Pr}\Big[\hat{H} \text{ is unacceptable}(3\,\Gamma + \epsilon)\text{-far}\Big]$$

$$= \mathbf{Pr}\big[\text{a random hypothesis in } Q \text{ is unacceptable}\big]$$

$$= \sum_{\ell=1}^{k+1} \frac{\mathbf{E}[\# \text{ unacceptable hypotheses in } Q \mid |Q| = \ell]}{\ell} \cdot \mathbf{Pr}[|Q| = \ell].$$

Now, we separate the first $t$ terms in the summation, and bound them from above by the probability of $|Q|$ being at most $t$. Moreover, we use the fact that the size of $|Q|$ is essentially a binomial random variable drawn from $\mathbf{Bin}(k, p_0)$ plus one. Hence, we obtain:

$$\mathbf{Pr}\Big[\hat{H} \text{ is } (3\,\Gamma + \epsilon)\text{-far}\Big] \le \mathbf{Pr}[|Q| \le t] + \sum_{\ell=t+1}^{k+1} \frac{\mathbf{E}[\# \text{ unacceptable hypotheses in } Q \mid |Q| = \ell]}{t} \cdot \mathbf{Pr}[|Q| = \ell]$$

$$\le \mathbf{Pr}[\mathbf{Bin}(k, p_0) < t] + \frac{\mathbf{E}[\# \text{ unacceptable hypotheses in } Q]}{t}$$

$$\le \mathbf{Pr}\left[\mathbf{Bin}(k, p_0) < \frac{k \cdot p_0}{2}\right] + \frac{1}{t}$$

$$\le \exp\left(-\frac{k \cdot p_0}{8}\right) + \frac{\delta}{2} \le \exp\left(-\frac{1}{2\,\delta}\right) + \frac{\delta}{2} \le \frac{3\,\delta}{4},$$

(By the Hoeffding inequality)

where the last inequality holds due to the fact that $e^{-1/(2\,\delta)} < \delta/4$ for $\delta \le 0.1$. Hence, the proof is complete. $\qquad\square$

### A.2 Proofs of Memory-data tradeoffs of random ladder tournament

**Lemma 3.1.** *Suppose we have $n$ hypotheses in $\mathcal{H}$. For every $p_0 \ge 1/n$, $\epsilon$, and an integer $t$ between two and $k = \Theta(1/p_0)$, one can implement Algorithm 2 in such away that it uses:*

$$s = O\left(\frac{1}{t\,p_0} \cdot \frac{\log p_0^{-1}}{\epsilon^2}\right) \text{ samples, and} \qquad b = O\left(t^2 \log\left(\frac{\log p_0^{-1}}{\epsilon}\right) + t \log n\right) \text{ bits of memory.}$$

*Proof.* Recall that it suffices to draw $m := O(\log p_0^{-1}/\epsilon^2)$ samples to make sure all the comparisons we make in Line 7 are accurate with probability at least 0.99. Since we obtain this accuracy bound via the union bound, we may reuse the samples. The idea here is to process every $t$ hypotheses together with the same set of $m$ samples. To implement this algorithm in $b$ bits of memory, we first draw $t$ random hypotheses and store their indices using $O(t \log n)$ bits. Then, we draw $m$ sample and store the Scheffé counts of every pair of hypotheses that are drawn. Counting the number of samples in each Scheffé sets takes $O(\log m) = O(\log(\log n/\epsilon))$ bits. And, there are $O(t^2)$ many of them. Now, we can make all the comparisons between the $t$ hypotheses using $O(t^2 \log m) = O(t^2 \log(\log p^{-1}/\epsilon))$ bits. Now, we run the first $t$ steps of Algorithm 2 and compute the current winner up to round $t$, $w_t$. Then, repeat this process for $w_t$ and the next $t - 1$ random hypotheses with fresh samples until we reach the end of $k$ hypotheses. Since at each round we process $t - 1$ hypotheses (with the exception of the first round), the number of rounds is $\lceil(k - 1)/(t - 1)\rceil = O(1/(t\,p_0))$, so we use $O(m/(t\,p_0))$ samples in total. Hence, Algorithm 2 uses $O(\log n/(t\epsilon^2 p_0))$ samples and $b = O(t^2 \log(\log n/\epsilon) + t \log n)$ bits of memory overall. $\qquad\square$

**Lemma 3.2.** *Suppose we have $n$ hypotheses in $\mathcal{H}$. For every $p_0 \ge 1/n$, $\epsilon$, and an integer $t$ between two and $k = \Theta(1/p_0)$, one can implement Algorithm 2 in such away that it uses:*

$$s = O\left(\frac{1}{t\,p_0} \cdot \frac{\log p_0^{-1}}{\epsilon^2}\right) \text{ samples, and} \qquad b = O\left(t \cdot \frac{\log(p_0^{-1}) \cdot \log n}{\epsilon^2} + t \log n\right) \text{ bits of memory.}$$

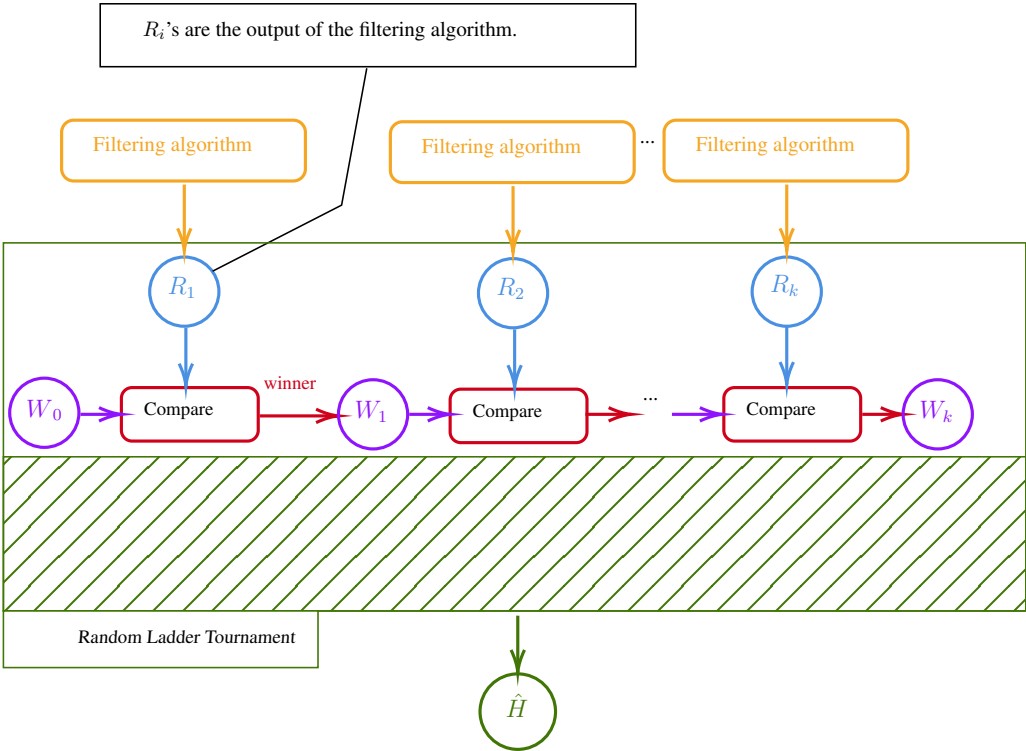

Figure 2: The sketch of our approach for the main result

*Proof.* The proof is very similar to the proof of Lemma 3.1. However, instead of using the Scheffé counts we use the sorted lists to store the samples. The only difference is that to process $t$ hypotheses, we will need $O(m\,t\,log n) = O(t\,\log p_0^{-1}\log n/\epsilon^2)$ bits instead of $O(t^2\log m)$ bits. This observation concludes the proof. $\square$

## B  Main result

In this section, we present an algorithm that achieves the tradeoff described in our main theorem. As we have described in Section 1.2, we obtain our main result by running the random ladder tournament on the hypotheses filtered by a filtering algorithm.

Recall that in the random ladder tournament, the algorithm can draw a random hypothesis such that with a probability of at least $p_0$, it is a $\Gamma$-close hypothesis. The number of comparisons made in the random ladder tournament is approximately $k := \Theta(1/p_0)$. To reduce $k$, we design a randomized filtering algorithm that produces an acceptable hypothesis $((3\,\Gamma + \epsilon')$-close to $P$) with a certain descent probability. Roughly speaking, this probability is $\min(1/\sqrt{n}, b/n)$. Although this probability is not very high, it is still greater than $1/n$, which allows us to execute the random ladder in fewer steps. The only caveat is that we must now run the random ladder tournament with a new parameter $\Gamma' := 3\,\Gamma + \epsilon'$. Hence, the accuracy guarantee of the resulting hypothesis will also be based on $\Gamma'$, and we will obtain a $(9\,\Gamma + \Theta(\epsilon'))$-close hypothesis to $P$. Figure 2 illustrates a visual representation of this approach.

The filtering algorithm operates on the basis of a simple observation: the excellent hypothesis $H^*$ will not lose to any unacceptable hypotheses (those that are $(3\,\Gamma + \epsilon')$-far). Therefore, if there is no decent hypothesis, one can identify the excellent hypothesis $H^*$ through an elimination tournament, where the algorithm pairs all hypotheses, compares them, and repeats this process while only the winners advance to the next round. Conversely, if numerous decent hypotheses exist, it suggests that a random hypothesis will have a reasonable chance of being decent (and hence acceptable). The

filtering algorithm combines these two strategies and does the following for each with probability a half:

1. It outputs a random hypothesis.

2. It runs an elimination tournament under the assumption that there are no decent hypotheses, then it outputs the result.

This approach leads to finding an excellent or decent hypothesis with an acceptable chance regardless of how scarce the decent hypotheses are.

Note that to run an elimination algorithm, we need to memorize the winners of each round. Hence, implementing the second part above with limited memory is not trivial. Therefore, we focus on a randomly sampled subset of elements and provide an algorithm that is extremely memory efficient. Roughly speaking, the algorithm processes $n$ hypotheses using only $O(n)$ bits of memory. This is one of the key components of our result that allows us to avoid extra $\log n$ factors and achieve the tradeoff $s \cdot b \approx O(n \log n)$.

The algorithm operates in rounds, employing a tree-like structure. In each round of this algorithm, we group a carefully chosen number of hypotheses and run an all-against-all tournament among them. Next, the winner of each group proceeds to the next round. We continue this process until we find a single winner.

In Section B.1, we describe the group elimination tournament. In Section B.2, we detail how to conduct the group elimination tournament on a random set of $n'$ hypotheses while utilizing only $n'$ bits of memory. In Section B.3, we outline the filtering algorithm. Finally, we conclude with Section B.4, which combines all the pieces together and presents the proof of our main result.

### B.1    Group elimination tournament

In this section, we focus on the case that there is no decent hypothesis in $\mathcal{H}$. That is, every hypothesis that is OPT-far from $P$ will lose to a hypothesis $H^*$ with $\|H^* - P\|_{\mathrm{TV}} = \mathsf{OPT}$. The main advantage of this assumption is that if we have a *knockout* style tournament: only an OPT-close hypothesis will survive to the top. Here, we propose an algorithm that implements a knockout style tournament, which we call group elimination tournament. However, instead of comparing two hypotheses at every step, we run the all-go-against-all tournament among a small group of hypotheses and send the winner to the next round. For simplicity, in this section, we describe our approach in Algorithm 3 where we assume no memory restriction. We prove the performance of this algorithm in Theorem 5. For a discussion on how to implement this algorithm using roughly $O(n)$ bits of memory, see Section B.2.

---

**Algorithm 3** Hypothesis selection with no decent hypothesis

---

1: **procedure** HYPOTHESIS-SELECTION-NO-DECENT($\mathcal{H}$, $n$, $\epsilon$, $\delta$, sample access to $P$)
2:     $\mathcal{H}_1 \leftarrow \mathcal{H}$
3:     **for** $\ell = 1, 2, \ldots, L$ **do**
4:         $\mathcal{G}_1, \mathcal{G}_2, \ldots, \mathcal{G}_{s_{\ell+1}} \leftarrow$ Partition $\mathcal{H}_\ell$ into groups of size $g_\ell$.
5:         $\mathcal{H}_{\ell+1} \leftarrow \emptyset$
6:         $S := \left\{ (i,j) \mid \exists \mathcal{G} \in \left\{ \mathcal{G}_1, \mathcal{G}_2, \ldots, \mathcal{G}_{s_{\ell+1}} \right\} \text{ such that } H_i \in \mathcal{G} \text{ and } H_j \in \mathcal{G} \right\}$
7:         Draw $m_\ell$ samples. For each sample, update the Scheffé counts of $\mathcal{S}(H_i, H_j)$ if $(i,j) \in S$
8:         **for** $\mathcal{G} = \mathcal{G}_1, \mathcal{G}_2, \ldots, \mathcal{G}_{s_{\ell+1}}$ **do**
9:             Run a ALL-GO-AGAINST-ALL TOURNAMENT($P, \mathcal{G}, \epsilon, \delta/(g_\ell)$) over the hypotheses in group $\mathcal{G}$.
10:             Add the winner to the set $\mathcal{H}_{\ell+1}$.
11:     Return the hypothesis in $\mathcal{H}_{L+1}$.

---

**Lemma B.1.** *Suppose we have a set of $n$ hypotheses $\mathcal{H} = \{H_1, H_2, \ldots, H_n\}$ and an unknown hypothesis $P$ which we wish to properly learn in $\mathcal{H}$. Suppose there are no decent hypothesis in $\mathcal{H}$, that is all hypotheses in $\mathcal{H}$ are either* OPT*-close, or they will lose to any* OPT*-close hypothesis. For every $2 \leq r \leq n$, there exists an algorithm that uses $O((\log(\log_r n) \cdot \log(1/\delta) + \log n)/\epsilon^2)$ samples from $P$, and it outputs $\hat{H}$ that is* OPT*-close to $P$ with probability $1 - \delta$.*

| Round # | # hypotheses | group size | # groups or # winners | space |
|---|---|---|---|---|
| 1 | $s_1 = n$ | $g_1 = \lceil \sqrt{r} \rceil$ | $s_1/g_1 \leq n/\sqrt{r}$ | $O(n \cdot r \cdot \log(\log(1/\delta)/\epsilon))$ |
| 2 | $s_2 = \lceil n/g_1 \rceil$ | $g_2 = \left\lceil (\sqrt{r})^{1.5} \right\rceil$ | $s_2/g_2 \leq n/r^{1.25}$ | $O(n \cdot r \cdot \log(\log(1/\delta)/\epsilon))$ |
| 3 | $s_3 = \lceil n/(g_1 g_2) \rceil$ | $g_3 = \left\lceil (\sqrt{r})^{2.25} \right\rceil$ | $s_3/g_3 \leq n/r^{2.375}$ | $O(n \cdot r \cdot \log(\log(1/\delta)/\epsilon))$ |
| | | $\vdots$ | | |
| $\ell = L$ | $s_\ell = \left\lceil \dfrac{n}{\prod_{\ell=1}^{L-1} g_\ell} \right\rceil$ | $g_\ell = \left\lceil (\sqrt{r})^{1.5^{\ell-1}} \right\rceil$ | $s_\ell/g_\ell \leq \dfrac{n}{r^{(1.5)^{\ell-1}}}$ | $O\left( \dfrac{s_\ell}{g_\ell} \cdot g_\ell^2 \cdot \log m_\ell \right) = O\left( n \cdot r \cdot \log\left( \dfrac{\log(1/\delta)}{\epsilon} \right) \right)$ |

Table 1: Branching factors in the tree structure

*Proof.* The algorithm runs in $L$ rounds. At round $\ell \in [L]$, we start with $s_\ell$ hypotheses. We partition them into groups of size $g_\ell$. For each round, we draw $m_\ell := \Theta(\log(g_\ell^3/\delta)/\epsilon^2)$ fresh samples. We keep track of the number of samples in the Scheffé sets of every pair of hypotheses that are in the same group. Using the Scheffé counts, we run an all-go-against-all tournament in each group $\mathcal{G}_i$ with overall confidence probability $1 - \delta/(6\,g_i)$ and find the winner. Then, we repeat this process among the winners in the next round until we end up with only one hypothesis. See Algorithm 3 for the description.

**Parameters:** We set the group sizes as follows:

$$g_1 = \left\lceil \sqrt{r} \right\rceil, \qquad g_\ell := \left\lceil (\sqrt{r})^{1.5^{\ell-1}} \right\rceil \approx g_{\ell-1}^{1.5} \qquad \forall \ell \in \{2, \ldots, L\}. \tag{2}$$

We start with $s_1 := n$. The number of hypotheses in the $\ell$-th round is:

$$s_\ell := \left\lceil \frac{s_{\ell-1}}{g_{\ell-1}} \right\rceil = \left\lceil \frac{n}{\prod_{i=1}^{\ell-1} g_i} \right\rceil \leq \left\lceil \frac{n}{r^{1.5^{\ell-1}-1}} \right\rceil \qquad \forall \ell \in \{2, \ldots, L\}. \tag{3}$$

The equation above is due to Fact D.2 and Fact D.3.

Observe that we stop our process when only one hypothesis is left: the ultimate winner. Thus, $L$ is the smallest integer such that $s_{L+1}$ is one. That is, $\prod_{\ell=1}^{L} g_\ell \geq n$. It is not hard to show that $L$ is $O\left( \log\left( \log_r n \right) \right)$ due to the following argument. For any $L \geq \log_{1.5}\left( \log_r n + 1 \right)$, one can show $\prod_{\ell=1}^{L} g_\ell$ is at least $n$ via Fact D.3:

$$\prod_{\ell=1}^{L} g_\ell \geq r^{1.5^L - 1} \geq n.$$

Therefore, $L$ cannot be larger than $\lceil \log_{1.5}\left( \log_r n + 1 \right) \rceil$.

We describe how we set our parameters in Table 1.

**Sample complexity:** For round $\ell$, we draw $m_\ell$ fresh samples. Thus, the total number of samples is:

$$\text{\# samples} = \sum_{\ell=1}^{L} m_\ell = \sum_{\ell=1}^{L} O\left(\frac{1}{\epsilon^2} \cdot \log \frac{g_\ell^3}{\delta}\right)$$

$$= O\left(\frac{L \cdot \log(1/\delta)}{\epsilon^2}\right) + O\left(\frac{1}{\epsilon^2}\right) \cdot \sum_{\ell=1}^{L} \log g_\ell$$

$$= O\left(\frac{L \cdot \log(1/\delta)}{\epsilon^2}\right) + O\left(\frac{1}{\epsilon^2}\right) \cdot \log \left(\prod_{\ell=1}^{L} g_\ell\right)$$

$$= O\left(\frac{L \cdot \log(1/\delta) + \log n}{\epsilon^2}\right) = O\left(\frac{\log(\log_r n) \cdot \log(1/\delta) + \log n}{\epsilon^2}\right).$$

Note that in the last line we use the fact that $\prod_{\ell=1}^{L} g_\ell$ is $\text{poly}(n)$ which is not difficult to prove using the following argument: We define $L$ to be the smallest integer such that $\prod_{\ell=1}^{L} g_\ell$ is at least $n$. Thus, $\prod_{\ell=1}^{L-1} g_\ell$ is at most $n$. Hence, we have:

$$\prod_{\ell=1}^{L} g_\ell = g_L \cdot \prod_{\ell=1}^{L-1} g_\ell \leq O(g_{L-1}^{1.5}) \cdot \prod_{\ell=1}^{L-1} g_\ell \leq O\left(\prod_{\ell=1}^{L-1} g_\ell\right)^{2.5} \leq O\left(n^{2.5}\right).$$

**Correctness:** Suppose $\mathcal{G}$ is a group in round $\ell$ that contains an OPT-close hypothesis. For the all-go-against-all tournament in round $\ell$ we use $m_\ell = O(\log(g_\ell^3/\delta)/\epsilon^2)$ samples. That implies we estimate the probability of all the Scheffé sets in $\mathcal{G}$ with accuracy $\epsilon$ with probability at least $1 - \delta/(6\,g_\ell)$. Given that we do not have any decent hypothesis, using Fact D.1, the all-go-against-all tournament has to output an OPT-close hypothesis with probability $1 - \delta/(6\,g_\ell)$.

Now, let $\mathcal{G}^{(1)}$ be the group in round one that $H^*$ belongs to. Let $\mathcal{G}^{(2)}$ be the group that the winner of $\mathcal{G}^{(1)}$ belongs to in round two, and similarly we define: $\mathcal{G}^{(3)}, \ldots, \mathcal{G}^{(L)}$. Clearly, the all-go-against-all tournament over $\mathcal{G}^{(i)}$ will return an OPT-close hypothesis with probability $1 - \delta/(6\,g_\ell)$ if $\mathcal{G}^{(i)}$ contains an OPT-close hypothesis. Thus, using the union bound, the probability that the final group $\mathcal{G}^{(L)}$ does not output an OPT-close hypothesis is bounded by the following:

$$\mathbf{Pr}[\text{Output hypothesis is not OPT} - \text{close}] \leq \sum_{\ell=1}^{L} \frac{\delta}{6\,g_\ell} \leq \frac{\delta}{6} \cdot \sum_{\ell=1}^{L} (\sqrt{r})^{-1.5^{\ell-1}} \leq \frac{\delta}{6} \cdot \sum_{\ell=1}^{L} (\sqrt{r})^{-\ell/2}$$

$$\leq \frac{\delta}{6} \cdot \frac{r^{-1/4}}{1 - r^{-1/4}} \leq \delta.$$

$\square$

### B.2 Memory usage of Algorithm 3

In this section, we provide an approach to implement Algorithm 3 using roughly $O(n)$ bits of memory and prove our main theorem.

**Theorem 5.** *Suppose we have a set of $n$ hypotheses $\mathcal{H} = \{H_1, H_2, \ldots, H_n\}$ and an unknown hypothesis $P$ which we wish to properly learn in $\mathcal{H}$. Suppose there are no decent hypothesis in $\mathcal{H}$, that is all hypotheses in $\mathcal{H}$ are either OPT-close, or they will lose to any OPT-close hypothesis. For every $2 \leq r \leq n$, there exists an algorithm that uses $O(n \cdot r \cdot \log(\log(1/\delta)/\epsilon))$ bits of memory and $O((\log(\log_r n) \cdot \log(1/\delta) + \log n)/\epsilon^2)$ samples from $P$, and it outputs $\hat{H}$ that is OPT-close to $P$ with probability $1 - \delta$.*

*Proof.* Note that there are two main sets of values that we need to store. First, the values of the Scheffé counts at every round that allow us to perform the all-go-against-all tournaments. Second, the

indices of the winner hypotheses that go to the next round. Below, we discuss bounding the memory usage for each part.

**Memory usage of Scheffé counts:** Here, we show to store the Scheffé counts, we need $O(n \cdot r \cdot \log(\log(1/\delta)/\epsilon))$ bits at every round. For a single group of size $g_\ell$, there are $O(g_\ell^2)$ pairs of hypotheses, therefore Scheffé sets, so we require $O(g_\ell^2 \log m_\ell)$ bits to count the number of samples in all the sets. Thus, in round $\ell$, we need the following number of bits:

$$
\begin{aligned}
\text{space} = \# \text{ groups} \cdot O(g_\ell^2 \cdot \log m_\ell) &= O\left(\frac{s_\ell}{g_\ell} \cdot g_\ell^2 \cdot \log m_\ell\right) \\
&= O\left(s_\ell \cdot g_\ell \cdot \left(\log g_\ell + \log\left(\frac{\log(1/\delta)}{\epsilon^2}\right)\right)\right) \\
&= n \cdot \log\left(\frac{\log(1/\delta)}{\epsilon}\right) \cdot O\left(\frac{s_\ell}{n} \cdot g_\ell \cdot (\log g_\ell)\right)
\end{aligned}
$$

Next, we show the last term in the last line is $O(r)$. For round 1, the memory bound holds since $\sqrt{r} \cdot \log(\sqrt{r}) \le r$. Now for $\ell \ge 2$ using Equation (3), we have:

$$
O\left(\frac{s_\ell}{n} \cdot g_\ell \cdot (\log g_\ell)\right) = O\left(\frac{g_\ell \cdot \log g_\ell}{r^{(1.5)^{\ell-1}-1}}\right) = O\left(\frac{r \cdot g_\ell \cdot \log g_\ell}{g_\ell^2}\right) = O(r).
$$

Hence, the bit complexity at every step is:

$$
O\left(n \cdot r \cdot \log\left(\frac{\log(1/\delta)}{\epsilon}\right)\right).
$$

**Storing the indices:** Note that to run the comparisons, one needs the true indices of the hypotheses to query the PDF-comparator or draw a sample. Thus, in the trivial implementation of our algorithm, we potentially need $O(n \log n)$ bits to keep track of the indices of the winners at every round. In this section, we explain how one can implement our algorithm in a memory-efficient manner, so keeping track of the winners' indices does not take more than $O(n)$ at every round.

We start by the following assumption that there is natural ordering among the hypotheses: $H_1, H_2, \ldots, H_n$. In this way, for the first round we do not need to remember the set of indices, we only need to remember $n$. At every level, we group "adjacent" hypotheses together and preserve this natural ordering. More precisely, at round one with group size $g_1$, the groups are:

$$
\begin{aligned}
\mathcal{G}_1 &:= \{H_1, H_2, \ldots, H_{g_1}\}, \\
\mathcal{G}_2 &:= \{H_{g_1+1}, H_{g_1+2}, \ldots, H_{2g_1}\}, \\
&\vdots \\
\mathcal{G}_{\lceil\frac{n}{g_1}\rceil} &:= \{H_{\lfloor\frac{n-1}{g_1}\rfloor \cdot g_1+1}, H_{\lfloor\frac{n-1}{g_1}\rfloor \cdot g_1+2}, \ldots, H_n\}.
\end{aligned}
$$

For the next round, we keep a list of $\lceil n/g_1 \rceil$ winners. However, instead of fully writing down the index of the winners, we use $O(\log(g_1))$ bits per each, and save the index of the winner within the group. For the next round, we partition the hypotheses into groups of size $g_2$ keeping the "adjacent" hypotheses in the same group. We repeat the same process to store the winners. However now, each winner are "representing" $O(g_1 \cdot g_2)$ hypotheses, so we need $O(\log(g_1 \cdot g_2))$ bits to store its index. We continue this approach. At round $\ell > 1$, we have a list of $s_\ell = O(n/\prod_i^{\ell-1} g_i)$ hypotheses, and we require $O(\log \prod_i^{\ell-1} g_i)$ many bits to do so. Thus, at every step we need $O(n)$ bits of memory.

Note that at every point in the process, it is easy to compute the actual index from the stored index. If right before starting round $i$, the $j$-th index in the list is $k$, then the true index is:

$$
\text{NEWINDEX}(i, j, k) = (j-1) \cdot \prod_i^{\ell-1} g_i + k.
$$

□

### B.3 Filtering acceptable hypotheses

In this section, we describe a filtering algorithm that allow us to pick an acceptable hypothesis with a modest chance. Our algorithm is simple we draw roughly $n' \ll n$ hypothesis at random. If one of these hypothesis is OPT-close, and there is no decent hypothesis among them, Algorithm 3 returns an excellent hypothesis. On the other hand, if there are a lot of decent hypotheses, then there is a modest chance that a randomly selected hypothesis is decent, hence an acceptable one.

In the description of Algorithm 3, we assume we have a class of hypotheses of size $n$, and roughly $O(n)$ bits of memory. However, we aim to use this algorithm in a setting that we can process $n'$ random hypotheses in roughly $O(n')$ bits of memory. The primary challenge here is that we need roughly $O(n' \log n)$ many bits to memorize which hypotheses are participating, and we cannot use the "natural ordering" of the hypotheses that we use in the proof of Theorem 5. Thus, we are looking for a mapping that maps indices in $[n']$ to a random set of indices in $[n]$ in a memory-efficient manner. To do so, we relax our requirement regarding that we need to draw $n'$ random hypotheses uniformly from $\mathcal{H}$. Instead, we are looking for a set of random pairwise independent indices while finding the index of the selected elements requires small amount of memory. This relaxation gives us a process that *filters* acceptable hypotheses:

**Theorem 6.** *Suppose we have a set of $n$ hypotheses $\mathcal{H} = \{H_1, H_2, \ldots, H_n\}$ and an unknown hypothesis $P$ which we wish to properly learn in $\mathcal{H}$. For every positive integer $n' < n$, there exists an algorithm that uses $O(n' \cdot \log(1/\epsilon))$ bits of memory and $O((\log n)/\epsilon^2)$ samples from $P$, and it outputs $\hat{H}$ that is excellent or decent with the probability:*

$$\mathbf{Pr}\Big[ \hat{H} \text{ is excellent or decent.} \Big] \geq \min\left( \frac{n'}{16\,n}, \frac{1}{4\,n'} \right) .$$

*Proof.* First, we start off by describing how we randomly pick $n'$ hypotheses from the set of $n$ hypotheses. We use a standard technique for generating pairwise independent random indices described in [Vad12, Chapter 3.5]. The following is the description of a randomized mapping that for any given index $x \in [n']$ it gives an index $i$ in $[n]$ that can be computed in a memory efficient manner.

Let $q$ be a prime number between $n$ and $2n$ ($n < q < 2n$). Such $q$ always exists via the Bertrand-Chebyshev theorem for any $n \geq 2$. Without loss of generality, we assume we have $q$ hypotheses by adding fake hypotheses to $\mathcal{H}$ that lose to any other hypothesis. Now, consider the finite field of $\mathbb{Z}_q$. Proposition 3.24 in [Vad12] implies that if we use two random numbers $a$ and $b$ in $\mathbb{Z}_q$, then the following mapping generates a set of pairwise independent indices:

$$f_{a,b}(x) := a \cdot x + b \,(\text{mod } q) .$$

More precisely, by iterating $x$ from 1 to $n'$, this randomized mapping selects $n'$ hypotheses with the following indices: $f_{a,b}(1), f_{a,b}(2), \ldots, f_{a,b}(n')$.[3] Note that to compute this mapping, we only need to memorize $a$ and $b$, which requires $O(\log n)$ bits, significantly smaller compared to $O(n' \log n)$ which we would need to memorize $n'$ random indices. Using the pairwise independence of the indices generated by this mapping, it is not too difficult to show that with some modest chance we pick exactly one excellent hypothesis and no decent hypothesis with this mapping.

**Lemma B.2.** *For a prime $q$, assume we have a class of $q$ hypotheses that contains $t_e \geq 1$ excellent hypotheses and $t_d$ decent hypotheses. Let $a \neq 0$ and $b$ be two random numbers in $\mathbb{Z}_q$ selected uniformly at random. Let $\mathcal{H}'$ be the set of following hypotheses: $\{H_{f_{a,b}(x)} : x \in [n']\}$ for which $f_{a,b}(x) := a \cdot x + b \,(\text{mod } q)$. If $(t_e + t_d)/q \leq 1/(2n')$, then the probability that $\mathcal{H}'$ contains exactly one excellent hypothesis and no decent hypothesis is at least $n' \cdot t_e/(2\,q)$.*

See Section D for the proof of this lemma.

---

[3] Without loss of generality, assume $H_0$ is $H_q$.

**Algorithm 4** Filtering Algorithm
______________________________________________________________________

1: **procedure** HYPOTHESES-FILTER($\mathcal{H}, n, \epsilon$, sample access to $P$)
2:      $C \leftarrow$ toss up a coin.
3:      **if** $C$ is head **then**
4:          Draw a random $i$ in $[n]$
5:          $\hat{H} \leftarrow H_i$
6:      **else**
7:          $q \leftarrow$ the first prime number that is at least $n$
8:          $a \leftarrow$ a random number in $[q-1]$
9:          $b \leftarrow$ a random number in $[q]$
10:        $\mathcal{H}' \leftarrow \{H_{a \cdot x + b \pmod q} \mid x \in [n']\}$
11:        $\hat{H} \leftarrow$ HYPOTHESIS-SELECTION-NO-DECENT$(\mathcal{H}', n', \epsilon, 0.5)$
12:      Output $\hat{H}$.
______________________________________________________________________

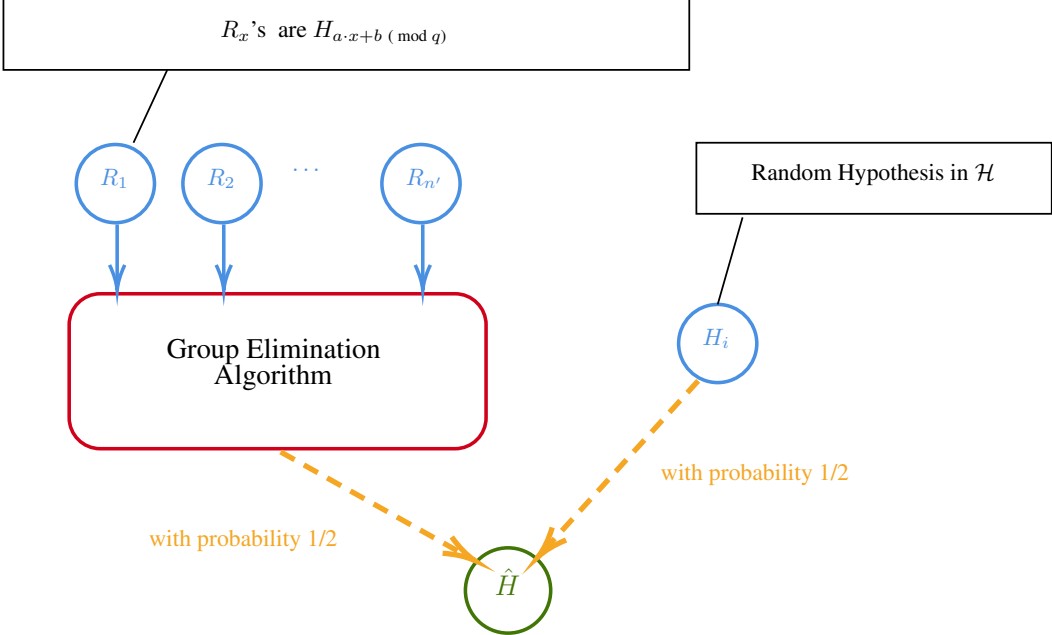

Figure 3: The sketch of the filtering algorithm

**The algorithm:** Assume $\mathcal{H}$ contains $t_e \geq 1$ prefect hypotheses and $t_d$ decent hypotheses. The algorithm is simple: with probability a half we set $\hat{H}$ to be a random hypothesis from $\mathcal{H}$. And, with probability a half we use the randomize mapping to pick $n'$ hypotheses and run Algorithm 3 with parameters $\delta = 0.5$ and $r = 2$ on them. And, we output the output of that algorithm. Figure 3 shows a visual representation of the filtering algorithm.

The probability of getting an acceptable hypothesis by picking a random one is $(t_e + t_d)/n$. Note that if this quantity is at least $1/2\, n'$, the statement of the lemma is true. Thus, assume $(t_e + t_d)/n$ is less than $1/2\, n'$ which implies that $(t_e + t_d)/q$ is less than $1/2\, n'$. Given this condition, by the above lemma, our randomized mapping selects a set of hypotheses with only one excellent hypothesis and no other acceptable hypotheses with probability at least $n' \cdot t_e/(2\, q)$. And, since we set $\delta$ to a half, with probability at least a half, Algorithm 3 will choose the excellent hypothesis as its output. Using that $q < 2\, n$ and $\mathbf{Pr}[C = \text{tail}] = 1/2$, with probability at least $n' \cdot t_e/(8\, q) \leq n'/(16\, n)$, $\hat{H}$ is the excellent hypothesis as desired in the statement. Hence, the proof is complete.     $\square$

### B.4 Putting everything together

**Theorem 1.** *There exists a constant $c$ for which the following holds. Let $\mathcal{H}$ be an arbitrary class consisting of $n$ distributions, and let $\epsilon > 0$. For every natural number $b$ where $c \cdot (\log n + \log((\log n)/\epsilon)) \le b \le n$, there exists an $(\alpha = 9, \epsilon, \delta = 0.1)$-proper learner (with promise) for $\mathcal{H}$ using $b$ bits of memory and the following number of samples:*

$$
s = \begin{cases}
O\left( \dfrac{n \log n}{b} \cdot \dfrac{1}{\epsilon^2} \log\left(\dfrac{1}{\epsilon}\right) \right) & \text{when } b \le \dfrac{n}{\log \log n}, \\[3mm]
O\left( \dfrac{n \log n}{b} \cdot \dfrac{1}{\epsilon^2} \log\left(\dfrac{\log n}{\epsilon}\right) \right) & \text{when } \dfrac{n}{\log \log n} \le b \le n.
\end{cases}
$$

*Proof.* Here, we combine the filtering algorithm and the random ladder tournament to prove the upper bound. We first use the filtering algorithm, Algorithm 4, to increase the probability of getting an acceptable hypothesis ($3\,\Gamma + \epsilon'$-close $P$) with parameter $\epsilon' = \epsilon/4$. Then, we use the random ladder tournament, Algorithm 2, to select an acceptable hypothesis among the hypotheses that have passed the filtering. Note that in the random ladder tournament, we need to use a new $\Gamma'$ parameter, that is, $3\,\Gamma + \epsilon'$.

Here, we use the memory bounds provided in Theorem 6 and Lemma 3.1 for these two algorithms. Fix two small constants $c_0, c_1 \le 1$ that we determine later. Let $t$ be the largest integer such that:

$$
t^2 \log\left(\frac{\log n}{\epsilon}\right) t + t \log n \le c_0 b.
$$

Note that given our lower bound for $b$, one can find a $t$ that is at least two. Set $n'$ as follows:

$$
n' := \min\left( \frac{c_1 \cdot b}{\log(1/\epsilon)}, 2\sqrt{n} \right).
$$

We consider two cases in the following:

**Case 1: $n' < 2\sqrt{n}$.** In this case, $n'$ is equal to $c_1 \cdot b / \log(1/\epsilon)$. One can run Algorithm 4 and find an acceptable hypothesis with probability $p_0 := \min(n'/(16\,n), 1/(4\,n')) = n'/(16\,n)$. Now in this case, we invoke Algorithm 4 $O(1/p_0)$ times, and in each round we use $O(\log n/\epsilon^2)$ samples. Furthermore, the random ladder tournament uses $O((\log p_0^{-1})/(t\,p_0\,\epsilon^2))$ samples. Therefore overall, our sample complexity is:

$$
O\left( \frac{1}{p_0} \cdot \left(1 + \frac{1}{t}\right) \cdot \frac{\log n}{\epsilon^2} \right) = O\left( \frac{1}{p_0} \cdot \frac{\log n}{\epsilon^2} \right) = O\left( \frac{n}{n'} \cdot \frac{\log n}{\epsilon^2} \right)
$$
$$
= O\left( \frac{1}{p_0} \cdot \frac{\log n}{\epsilon^2} \right) = O\left( \frac{n \log n}{b\,\epsilon^2} \cdot \log \frac{1}{\epsilon} \right).
$$

**Case 2: $n' = 2\sqrt{n}$.** Similar to the previous case, we would like to run Algorithm 4 and find an acceptable hypothesis with probability $p_0 := \min(n'/(16\,n), 1/(4\,n')) = 1/(4\,n')$. Now, since $n' \le c_1 \cdot b / \log(1/\epsilon)$, we may be able to run multiple instances of Algorithm 4 in parallel while using the same number of samples. Let $r = c_1 \cdot b / (n' \cdot \log(1/\epsilon))$. Now, the number of samples we use for this stage is:

$$
O\left( \frac{1}{r\,p_0} \cdot \frac{\log n}{\epsilon^2} \right) = O\left( \frac{n'}{r} \cdot \frac{\log n}{\epsilon^2} \right) = O\left( \frac{\log(1/\epsilon) \cdot \log n}{b\,\epsilon^2} \right).
$$

Now, we focus on the number of samples we need for the random ladder tournament. Note that $t$ should be:

$$t = \Theta\left(\min\left(\frac{b}{\log n}, \sqrt{\frac{b}{\log\left((\log n)/\epsilon\right)}}\right)\right).$$

Thus, we use the following amount of samples:

$$O\left(\frac{1}{t\,p_0} \cdot \frac{\log n}{\epsilon^2}\right) = O\left(\frac{\sqrt{n}}{b \cdot \min\left((\log n)^{-1}, \left(\sqrt{b} \cdot \log\left((\log n)/\epsilon\right)\right)^{-1}\right)} \cdot \frac{\log n}{\epsilon^2}\right)$$

$$= O\left(\frac{\max\left(\log n, \sqrt{b}\log\left((\log n)/\epsilon\right)\right)}{\sqrt{n}} \cdot \frac{n\,\log n}{b\,\epsilon^2}\right) = O\left(\frac{n\,\log n}{b\,\epsilon^2}\log\left(\frac{\log n}{\epsilon}\right)\right)$$

Putting these two cases together, we obtain the desire sample-memory tradeoffs:

$$\#\text{samples} \cdot \#\text{bits} = O\left(\frac{n\,\log n}{\epsilon^2}\log\left(\frac{\log n}{\epsilon^2}\right)\right).$$

Regarding the accuracy guarantee, it is not hard to see that in the filtration algorithm, with a modest probability, we pick a distribution that is $(3\,\Gamma + \epsilon')$-close to $P$. Now, we run the random ladder tournament with a new $\Gamma' = 3\,\Gamma + \epsilon'$. Then, we get a hypothesis with accuracy:

$$3\,\Gamma' + \epsilon' = 3\,(3\,\Gamma + \epsilon') + \epsilon' = 9\,\Gamma + 4\epsilon' = 9\,\Gamma + \epsilon.$$

Hence, the proof is complete. $\qquad\square$

## C   Proper learner without promise

Throughout this paper, we mainly focus on proper learners with promise, for which the algorithm is given a parameter $\Gamma$, and we are promised that $\mathsf{OPT}$ is at most $\Gamma$. In this section, we focus on the case where no such information is available to the algorithm. First, we formally define a proper learner without promise.

**Definition C.1.** *Suppose $\mathcal{A}$ has sample access to an unknown distribution $P$. And, it can query the probabilities of the Scheffé sets according to each $H_i \in \mathcal{H}$ and a PDF-comparator for every pair of hypotheses in $\mathcal{H}$. We say algorithm $\mathcal{A}$ is an $(\alpha, \epsilon, \delta)$-proper learner for $\mathcal{H}$ if the following holds: for every distribution $P$, and for every class of $n$ distributions $\mathcal{H}$, with probability at least $1 - \delta$ over its input sample (drawn i.i.d. from $P$) and internal coin tosses, $\mathcal{A}$ outputs a distribution $\hat{H} \in \mathcal{H}$ such that:*

$$\|P - \hat{H}\|_{TV} \le \alpha \cdot \mathsf{OPT} + \epsilon.$$

Here, we provide a reduction from a proper learner without promise to a proper learner with promise. At a high level, we perform a binary search over possible values of $\Gamma$ in $(0, 1)$. At every step, we try a value of $\Gamma_m$ and run $\mathcal{A}$ with $\Gamma = \Gamma_m$. The main challenge is that regardless of $\Gamma_m$ being at least $\mathsf{OPT}$ or not, $\mathcal{A}$ may return a hypothesis that seems close to $P$, and it is hard to refute that the suggested value of $\Gamma$ is at least $\mathsf{OPT}$. For this reason, we look into the output of $\mathcal{A}$, say $\tilde{H}$, and see if we find evidence that $\tilde{H}$ is far from $P$. We check the distance between $\tilde{H}$ and $P$ on every Scheffé set of $\tilde{H}$. More precisely, we check if $W(\tilde{H})$ is larger than $(\alpha \cdot \mathsf{OPT} + \epsilon)$ where $W(H)$ is defined as follows for every hypothesis $H_i$:

$$W(H_i) := \max_{j \in [n]\setminus\{i\}} |H_i\left(\mathcal{S}(H_i, H_j)\right) - P\left(\mathcal{S}(H_i, H_j)\right)|.$$

Now, if $W(\tilde{H})$ is larger than $(\alpha \cdot \mathsf{OPT} + \epsilon)$, then one can imply that $\tilde{H}$ is not $(\alpha \cdot \mathsf{OPT} + \epsilon)$-close to $P$, therefore, $\Gamma_m$ must have been less than $\mathsf{OPT}$. On the other hand, if $W(\tilde{H})$ is at most $(\alpha \cdot \mathsf{OPT} + \epsilon)$, one can show $\tilde{H}$ is not too far away from $P$ (even when $\Gamma_m$ is less than $\mathsf{OPT}$). Putting these observations together and accounting for errors in estimation of $W(\tilde{H})$, we obtain a reduction that is described in Algorithm 5, and we prove its accuracy in Theorem 7.

---

**Algorithm 5** Reduction from a proper learner without promise to a proper learner with promise

1: **procedure** $\mathcal{B}(\mathcal{H}, \alpha, \epsilon, \epsilon', \delta, \delta',$ sample access to $P$, and oracle access to $\mathcal{A})$
2:     $\Gamma_\ell \leftarrow 0$
3:     $\Gamma_h \leftarrow 1$
4:     $\hat{H} \leftarrow \mathcal{A}(\mathcal{H}, \Gamma_h, \epsilon, \delta)$
5:     **while** $\Gamma_h - \Gamma_\ell > \epsilon$ **do**
6:         $\Gamma_m \leftarrow \dfrac{\Gamma_h + \Gamma_\ell}{2}$
7:         $\tilde{H} \leftarrow \mathcal{A}(\mathcal{H}, \Gamma_m, \epsilon, \delta)$
8:         $\tilde{W}(\tilde{H}) \leftarrow$ Estimate $W(\tilde{H})$ with error at most $\epsilon'$ and with probability at least $1 - \delta'$.
9:         **if** $\tilde{W}(\tilde{H}) \leq \alpha \Gamma_m + \epsilon + \epsilon'$ **then**
10:             $\Gamma_h \leftarrow \Gamma_m$
11:             $\hat{H} \leftarrow \tilde{H}$
12:         **else**
13:             $\Gamma_\ell \leftarrow \Gamma_m$
14:     Output $\hat{H}$ .

---

**Theorem 7.** *Fix five parameters $\epsilon, \epsilon', \delta, \delta' \in (0,1)$, and $\alpha \geq 1$. Assume that $\mathcal{A}$ is an $(\alpha, \epsilon, \delta)$- proper learner (with promise). Suppose that we can estimate the probabilities of all the Scheffé sets according to $P$ with additive error at most $\epsilon'$ and with probability at least $1 - \delta'$. Then, $\mathcal{B}$, described in Algorithm 5, is an $(\alpha + 2, (\alpha + 1)\epsilon + 2\epsilon', t \cdot \delta + \delta')$ proper learner (without promise) where $t$ is defined as $\lceil \log_2(1/\epsilon) \rceil + 1$.*

*Proof.* Our goal here is to show that for the output hypothesis $\hat{H}$ with probability at least $1 - t \cdot \delta + \delta'$, we have:

$$\|\hat{H} - P\|_{\mathrm{TV}} \leq (\alpha + 2)\,\mathsf{OPT} + (\alpha + 1)\epsilon + 2\,\epsilon' . \tag{4}$$

Observe that at every iteration of the while loop in the algorithm, $\Gamma_h - \Gamma_\ell$ is divided by two. Thus, the while loop takes $\lceil \log_2(1/\epsilon) \rceil$ iterations. Thus, $t$ is the number of times that we invoke $\mathcal{A}$ in Algorithm 5. Each time $\mathcal{A}$ works as expected with probability $1 - \delta$. Hence, using the union bound, with probability $1 - t \cdot \delta$, we can assume all invocations of $\mathcal{A}$ result in correct answers. In addition, with probability $1 - \delta'$, all the estimates we use for $P\left(\mathcal{S}(\hat{H}, H_i)\right)$ in Line 8 are accurate up to an additive error $\epsilon'$. Hence, $\left|\tilde{W}(\tilde{H}) - W(\tilde{H})\right|$ is at most $\epsilon'$ with probability at least $1 - \delta'$. Hence, by the union bound, the following event happens with probability at lest $1 - (t \cdot \delta + \delta')$: In every invocation of $\mathcal{A}$, if $\mathsf{OPT}$ is below the given parameter $\Gamma$, then $\mathcal{A}$ returns a hypothesis, $\tilde{H}$, that is $(\alpha\Gamma + \epsilon)$-close to $P$. And, $\tilde{W}(\tilde{H})$ is at most $\alpha\Gamma + \epsilon + \epsilon'$. Hence, for the rest of this proof, we assume this event holds, and we prove Equation 4.

Consider $\Gamma_h$ and $\Gamma_\ell$ in the final iteration of while loop in Algorithm 5. Observe that the output $\hat{H}$ is what $\mathcal{A}$ has found when it was invoked with input $\Gamma = \Gamma_h$. Now, we consider two cases as follows:

**Case 1: $\mathsf{OPT} \leq \Gamma_h$:**   In this case, $\hat{H}$ is $(\alpha \cdot \Gamma_h + \epsilon)$-close to $P$ by our assumption that $\mathcal{A}$ works as expected. Thus, $\|\hat{H} - P\|_{\mathrm{TV}}$ is at most $\alpha \cdot \Gamma_h + \epsilon$. This upper bound is written based on $\Gamma_h$, while we aim to find an upper bound on the distance that only contains $\mathsf{OPT}$. For this purpose, we need to use the fact that why the binary search algorithm stoped at this particular $\Gamma_h$. For $\Gamma_\ell$, we have two possibilities: either $\Gamma_\ell$ is zero; Or, we have run $\mathcal{A}$ with $\Gamma = \Gamma_\ell$, and we have not found a desired $\tilde{H}$.

In both cases, $\Gamma_\ell$ is a lower bound on OPT. Thus, we have:

$$\mathsf{OPT} \geq \Gamma_\ell \geq \Gamma_h - \epsilon \,.$$

Now, using this relationship between $\Gamma_h$ and OPT, we obtain:

$$\|\hat{H} - P\|_{\mathrm{TV}} \leq \alpha \cdot \Gamma_h + \epsilon \leq \alpha \cdot \mathsf{OPT} + (\alpha + 1)\,\epsilon \,,$$

implying Equation 4.

**Case 2: $\mathsf{OPT} > \Gamma_h$:** In this case, clearly, $\Gamma_h$ cannot be one (since OPT is at most one). Thus, $\Gamma_h$ was set in Line 10 implying that $\tilde{W}(\hat{H})$ is at most $\alpha\,\Gamma_h + \epsilon + \epsilon'$. Since we assumed the estimation of $W(\hat{H})$ was accurate up to error $\epsilon'$, we have:

$$W(\hat{H}) \leq \tilde{W}(\hat{H}) + \epsilon' \leq \alpha\,\Gamma_h + \epsilon + 2\,\epsilon' \,.$$

Given this bound on $W(\hat{H})$, we show that $\hat{H}$ cannot be far from $P$. Let $H^*$ be a hypothesis in $\mathcal{H}$ that is OPT-close to $P$. Then, we use the definition of the Scheffé sets and the triangle inequality to bound the distance between $\hat{H}$ and $P$:

$$
\begin{aligned}
\|\hat{H} - P\|_{\mathrm{TV}} &\leq \|\hat{H} - H^*\|_{\mathrm{TV}} + \|H^* - P\|_{\mathrm{TV}} && \text{(by triangle inequality)} \\
&= \left| \hat{H}\left(\mathcal{S}\left(\hat{H}, H^*\right)\right) - H^*\left(\mathcal{S}\left(\hat{H}, H^*\right)\right) \right| + \|H^* - P\|_{\mathrm{TV}} && \text{(by dfn. of Scheffé set)} \\
&\leq \left| \hat{H}\left(\mathcal{S}\left(\hat{H}, H^*\right)\right) - P\left(\mathcal{S}\left(\hat{H}, H^*\right)\right) \right| \\
&\quad + \left| P\left(\mathcal{S}\left(\hat{H}, H^*\right)\right) - H^*\left(\mathcal{S}\left(\hat{H}, H^*\right)\right) \right| + \|H^* - P\|_{\mathrm{TV}} && \text{(by triangle inequality)} \\
&\leq \left| \hat{H}\left(\mathcal{S}\left(\hat{H}, H^*\right)\right) - P\left(\mathcal{S}\left(\hat{H}, H^*\right)\right) \right| + 2\,\|H^* - P\|_{\mathrm{TV}} \,.
\end{aligned}
$$

Now, we use the definition of $W(\hat{H})$ and conclude:

$$
\begin{aligned}
\|\hat{H} - P\|_{\mathrm{TV}} &\leq W(\hat{H}) + 2\,\|H^* - P\|_{\mathrm{TV}} = W(\hat{H}) + 2\,\mathsf{OPT} \\
&\leq \alpha\,\Gamma_h + \epsilon + 2\,\epsilon' + 2\,\mathsf{OPT} \leq (\alpha + 2)\,\mathsf{OPT} + \epsilon + 2\,\epsilon' \,.
\end{aligned}
$$

Therefore, the proof is complete. $\square$

**Corollary C.2.** *For every parameters $\epsilon''$ and $\delta''$ in $(0,1)$, there exists an $(\alpha = 5, \epsilon = \epsilon'', \delta = \delta'')$-proper learner (without promise) that uses $O\left(\log(n/\delta'')/(\epsilon'')^2\right)$ samples and runs $\tilde{O}\left(n/\left(\delta''\,(\epsilon'')^2\right)\right)$ time.*

*Proof.* First, we start off with the parameters: We set $\epsilon := \epsilon''/6$, $\epsilon' := \epsilon''/6$, $\delta' = \delta''/2$, and $\delta = \delta''/(2\,t)$ where $t := \lceil \log_2(1/\epsilon) \rceil + 1$ is the number of iterations in Algorithm 5. Using the Chernoff bound and reusing all samples, we can estimate all the Scheffé sets with additive error at most $\epsilon'$ with probability at least $1 - \delta'$.

Note that based on Corollary A.1, the random ladder tournament (with an arbitrarily small confidence parameter) is a $(\alpha = 3, \epsilon, \delta)$-proper learner (with promise) that uses $O\left(\log(n/\delta)/\epsilon^2\right)$ samples and runs in $O\left(n\,\log(n/\delta'')/\left(\delta''\,(\epsilon'')^2\right)\right)$ time. We use this learner as our $\mathcal{A}$ and run the reduction algorithm $\mathcal{B}$ with the parameters we specified earlier. Now, using Theorem 7, we obtain a proper learner without promise that satisfies the following guarantee with probability at least $1 - \delta'' = 1 - (\delta' + t \cdot \delta)$:

$$\|\hat{H} - P\|_{\mathrm{TV}} \leq (\alpha + 2) \cdot \mathsf{OPT} + (\alpha + 1)\epsilon + 2\,\epsilon' = 5 \cdot \mathsf{OPT} + \epsilon'' \,.$$

Regarding the time complexity, the while loop in Algorithm 5 is repeated $t$ times. In each iteration, we compute $W(\tilde{H})$ which takes $O\left(n \log\left(n/\delta''\right)/\left(\epsilon''\right)^2\right)$ time, and we call the random ladder tournament (i.e, $\mathcal{A}$). Hence, the overall time complexity is:

$$O\left(t \cdot \frac{n \log\left(n/\delta''\right)}{\delta''\left(\epsilon''\right)^2}\right) = O\left(\frac{n \cdot \log(n/\delta'') \cdot \log\left(1/\epsilon''\right)}{\delta''\left(\epsilon''\right)^2}\right) = \tilde{O}\left(\frac{n}{\delta''\left(\epsilon''\right)^2}\right).$$

Hence, the proof is complete. $\qquad\square$

**Memory-data consumption of reduction algorithm:** Besides the sample and memory usage of $\mathcal{A}$, $\mathcal{B}$ needs to keep track of $O(1)$ numbers and also computes $\tilde{W}(\tilde{H})$. Now, given $b$ bits of memory, we can partition $\mathcal{H}$ into consecutive blocks of size $r = b/(\log(\log(n)/\epsilon))$. At every round, we process one block of size $r$, we keep track of the Scheffé counts of $\tilde{H}$ and the hypotheses in the block. For each round we use $O(\log(n)/\epsilon^2)$ samples, ane we have $O(n/r)$ rounds. Thus, the total number of samples is $s := O(n/r \cdot (\log(n)/\epsilon^2))$. Thus, for this part we will have:

$$s \cdot b = O\left(\frac{n \log n}{\epsilon^2} \cdot \log\left(\frac{\log n}{\epsilon}\right)\right).$$

**Remark 8.** *Using the reduction algorithm with the memory-data tradeoff described above, one can translate our main theorem, Theorem 1, to a $(\alpha = 11, \epsilon, 0.1)$-proper learner (without promise) incurring additional factors of* $\mathrm{poly}\left(\log\log(n), \log(1/\epsilon)\right)$.

# D   Auxiliary lemmas and facts

**Fact D.1.** *[adapted from Theorem 4 in [MS08]] For every pair $i, j \in [n]$, let $q_{ij}$ denote an estimate of the probabilities of the Scheffé set of $H_i$ and $H_j$ according to $P$. Let $\hat{H}$ be the hypothesis that minimizes of the following:*

$$\hat{H} := \arg\max_{H_i \in \mathcal{H}} \min_{j \in [n]\setminus\{i\}} \left|H_i\left(\mathcal{S}(H_i, H_j)\right) - q_{ij}\right|.$$

*For every $\epsilon$ and $\delta$, if we estimate $q_{ij}$ using $O(\log(n/\delta)/\epsilon^2)$ samples from $P$, then $\hat{H}$ satisfies the following guarantee:*

$$\|\hat{H} - X\|_{TV} \leq 3\,\mathsf{OPT} + \epsilon.$$

**Lemma 2.1.** *Upon receiving three parameters: $\Gamma$, $\epsilon$, and $\delta$, Algorithm 1 uses $O(\log(1/\delta)/\epsilon^2)$ samples and satisfies the following guarantees with probability at least $1 - \delta$:*

- *If $H_1$ is $\Gamma$-close to $P$, then the algorithm returns $H_1$.*

- *If $H_1$ is $(\max\left(\Gamma, \|H_2 - P\|_{TV}\right) + 2\|H_2 - P\|_{TV} + \epsilon)$-far from $P$, then the algorithm returns $H_2$.*

*Proof.* Let $q$, $p_1$, and $p_2$ denote the true probability of the Scheffé set of $H_1$ and $H_2$ according to $P$, $H_1$, and $H_2$:

$$q := P(\mathcal{S}(H_1, H_2)), \qquad p_1 := H_1(\mathcal{S}(H_1, H_2)), \qquad p_2 := H_2(\mathcal{S}(H_1, H_2)).$$

And, let $\hat{q}$ be the estimate of $q$ from the samples. the Using the Hoeffding bound, one can show that $|q - \hat{q}|$ is at most $\epsilon' := \epsilon/2$ with probability $1 - \delta$, since we use $O(\log(1/\delta)/\epsilon^2)$ samples. Now, if $\|H_1 - P\|_{TV}$ is at most $\Gamma$, then $|p_1 - q|$ must be at most $\Gamma$ implying:

$$|p_1 - \hat{q}| \leq |p_1 - q| + |q - \hat{q}| \leq \Gamma + \epsilon'.$$

Therefore, the algorithm outputs $H_1$ as the winner.

Now, we show the second guarantee. If the total variation distance between $H_1$ and $P$, $\|H_1 - P\|_{\text{TV}}$, is greater than $(\max(\Gamma, \|H_2 - X\|_{\text{TV}}) + 2\|H_2 - X\|_{\text{TV}} + \epsilon)$, then by the triangle inequality and the definition of total variation distance, we will have:

$$
\begin{aligned}
|p_1 - \hat{q}| \geq |p_1 - p_2| - |p_2 - \hat{q}| &\geq |p_1 - p_2| - |p_2 - q| - |q - \hat{q}| \\
&\geq \|H_1 - H_2\|_{\text{TV}} - \|H_2 - P\|_{\text{TV}} - \epsilon' \quad \text{(By the definition of the Scheffé set } \mathcal{S}(H_1, H_2)) \\
&\geq \|H_1 - P\|_{\text{TV}} - 2\|H_2 - P\|_{\text{TV}} - \epsilon' \\
&> \max(\Gamma, \|H_2 - X\|_{\text{TV}}) + \epsilon - \epsilon' \\
&\geq \max(\Gamma + \epsilon', |p_2 - q| + \epsilon') \\
&\geq \max(\Gamma + \epsilon', |p_2 - \hat{q}| + |q - \hat{q}|) \\
&\geq \max(\Gamma + \epsilon', |p_2 - \hat{q}|)
\end{aligned}
$$

Thus, the algorithm outputs $H_2$ as the winner. $\qquad\square$

**Lemma B.2.** *For a prime $q$, assume we have a class of $q$ hypotheses that contains $t_e \geq 1$ excellent hypotheses and $t_d$ decent hypotheses. Let $a \neq 0$ and $b$ be two random numbers in $\mathbb{Z}_q$ selected uniformly at random. Let $\mathcal{H}'$ be the set of following hypotheses: $\{H_{f_{a,b}(x)} : x \in [n']\}$ for which $f_{a,b}(x) := a \cdot x + b \pmod{q}$. If $(t_e + t_d)/q \leq 1/(2n')$, then the probability that $\mathcal{H}'$ contains exactly one excellent hypothesis and no decent hypothesis is at least $n' \cdot t_e/(2q)$.*

*Proof.* Using Proposition 3.24 in [Vad12] the mapping is in fact a class of pairwise independent hash functions. That is, the following holds:

- Fix two indices $i \in \mathbb{Z}_p$ and $x \in [n']$. The probability of mapping $x$ to $i$ is:
$$
\mathbf{Pr}_{a,b}[f_{a,b}(x) = i] = \frac{1}{q}.
$$

- Fix four indices $i, j \in \mathbb{Z}_p$ and $x, y \in [n']$ where $x \neq y$. The probability of mapping $x$ to $i$ and $y$ to $j$ is:
$$
\mathbf{Pr}_{a,b}[f_{a,b}(x) = i \text{ and } f_{a,b}(y) = j] = \frac{1}{q^2}
$$

Suppose $\mathcal{H}_e$ and $\mathcal{H}_d$ indicate the set of excellent and decent hypotheses respectively. Fix an index $i$ for which $H_i$ is an excellent hypothesis. Suppose one of the indices in $[n']$, say $x$, is mapped to $i$. Now, the probability of another hypothesis in $\mathcal{H}_e \cup \mathcal{H}_d$ being chosen for a fixed index $y \neq x$ in $[n']$ is:

$$
\begin{aligned}
\mathbf{Pr}_{a,b}\big[H_{f_{a,b}(y)(\text{mod } q)} \in \mathcal{H}_e \cup \mathcal{H}_d \mid f_{a,b}(x) = i\big] &= \sum_{j \text{ s.t. } H_j \in ((\mathcal{H}_e \cup \mathcal{H}_d)\setminus\{H_i\})} \mathbf{Pr}_{a,b}[f_{a,b}(y) = j \mid f_{a,b}(x) = i] \\
&= \sum_{j \text{ s.t. } H_j \in ((\mathcal{H}_e \cup \mathcal{H}_d)\setminus\{H_i\})} \frac{\mathbf{Pr}_{a,b}[f_{a,b}(y) = j \text{ and } f_{a,b}(x) = i]}{\mathbf{Pr}_{a,b}[f_{a,b}(x) = i]} \\
&= \frac{t_e + t_d - 1}{q}.
\end{aligned}
$$

Hence, the expected number of hypotheses (other than $H_i$) that are selected is:

$$
\begin{aligned}
\mathbf{E}\big[\#y \in [n'] \setminus \{x\} \text{ s.t. } H_{f_{a,b}(y)(\text{mod } q)} \in \mathcal{H}_e \cup \mathcal{H}_d \mid f_{a,b}(x) = i\big] &= \frac{(n'-1) \cdot (t_e + t_d - 1)}{q} \\
&\leq \frac{n' \cdot (t_e + t_d)}{q} \leq \frac{1}{2}.
\end{aligned}
$$

The last inequality is due to the assumption we have: $(t_e + t_d)/q \leq 1/(2n')$. Next, by Moarkov's inequality, the probability that the number of $y$'s are at least one (that is two times the expected value) is at most $1/2$. Hence, if $x$ is mapped to $H_i$, with probability at least $1/2$, there is no other excellent or decent hypothesis is selected.

Now, we compute the probability that an excellent hypothesis is selected and no other excellent or decent hypothesis is selected as follows:

$\mathbf{Pr}_{a,b}[\text{One excellent hypothesis and no decent one}]$

$$= \sum_{x \in [n']} \sum_{H_i \in \mathcal{H}_e} \mathbf{Pr}[f_{a,b}(x) = i \pmod{q} \text{ and no other hypothesis in } \mathcal{H}_e \cup \mathcal{H}_d \text{is selected}]$$

$$= \sum_{x \in [n']} \sum_{H_i \in \mathcal{H}_e} \mathbf{Pr}[f_{a,b}(x) = i \pmod{q}]$$

$$\cdot \mathbf{Pr}\Big[\text{no } y \in [n'] \setminus \{x\} \text{ s.t. } H_{f_{a,b}(y) \pmod{q}} \in \mathcal{H}_e \cup \mathcal{H}_d \mid f_{a,b}(x) = i \pmod{q}\Big]$$

$$\geq \sum_{x \in [n']} \sum_{H_i \in \mathcal{H}_e} \mathbf{Pr}[f_{a,b}(x) = i \pmod{q}] \cdot \frac{1}{2} = \frac{n' \cdot t_e}{2\,q}$$

Hence, the proof is complete. $\qquad\square$

**Fact D.2.** *Let $n, a, b$ be three positive integers. Then, we have:*

$$\left\lceil \frac{\lceil n/a \rceil}{b} \right\rceil = \left\lceil \frac{n}{a\,b} \right\rceil .$$

*Proof.* Let $r$ denote the remainder of $n$ divided by $a$. If $r = 0$, then the claim is trivial. Therefore, assume $r$ is in $\{1, 2, \ldots, a-1\}$. Let $s$ be the remainder of $n$ divided by $a\,b$. Clearly, $s$ has to be of the form: $t \cdot a + r$ for $t \in \{0, 1, \ldots, b-1\}$. Hence, we have

$$\left\lceil \frac{\lceil n/a \rceil}{b} \right\rceil = \left\lceil \frac{n}{a\,b} + \frac{a-r}{a\,b} \right\rceil = \left\lfloor \frac{n}{a\,b} \right\rfloor + \left\lceil \frac{s+a-r}{a\,b} \right\rceil$$

$$= \left\lfloor \frac{n}{a\,b} \right\rfloor + \left\lceil \frac{(t+1) \cdot a}{a\,b} \right\rceil = \left\lfloor \frac{n}{a\,b} \right\rfloor + 1 = \left\lceil \frac{n}{a\,b} \right\rceil .$$

$\qquad\square$

**Fact D.3.** *Given the definition of the $g_\ell$'s in Equation (2), for any positive integer $\ell$, we have:*

$$\prod_{i=1}^{\ell} g_i \geq r^{1.5^\ell - 1} .$$

*Proof.* By the properties of the geometric series, we obtain:

$$\prod_{i=1}^{\ell} g_i \geq \prod_{i=1}^{\ell} \left(\sqrt{r}\right)^{1.5^{i-1}} = \left(\sqrt{r}\right)^{\sum_{i=1}^{\ell} 1.5^{i-1}} = \left(\sqrt{r}\right)^{(1.5^\ell - 1)/(1.5 - 1)} = r^{1.5^\ell - 1} .$$

$\qquad\square$

