# Hypothesis Selection with Memory Constraints

Maryam Aliakbarpour [*]
Rice University
maryama@rice.edu

Mark Bun [†]
Boston University
mbun@bu.edu

Adam Smith [‡]
Boston University
ads22@bu.edu

(Full version)

October 27, 2023

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

**An equivalent model: limited communication on a chain.** Although we present our results in terms of a limited-memory learner, one may also view the model as a distributed one, in which each sample lives on a separate machine and machines are arranged in a chain: messages are passed from one machine to the next until they reach the final one, which must produce an output. The memory bound $b$ corresponds to the maximum size of a message in such a model, while the sample size $s$ corresponds to the number of machines. Such a limited-communication model makes sense when the samples are themselves such large objects that communicating the raw data would be prohibitively expensive.

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

}|$. They showed that a constant $\alpha = 2$ is achievable for improper learning. This result was then improved in [BBK+21], which achieved the constant 2 with nearly optimal sample complexity.

Aamand et al. in [AAC+23] also study statistical computational tradeoffs (time-data) for the hypotheses over *discrete domains*. Unlike our work, their setting allows pre-processing of the class $\mathcal{H}$ in polynomial time. They create a data structure that runs in sub-linear time in $n$ while the sample complexity depends on the domains size of the distribution (sublinear in $|\mathcal{X}|$). It is worth noting that the time and sample complexities of our algorithms are independent of the domain size, allowing us to work with both discrete and continuous distributions alike.

There is a long history of research focusing on a special case of this problem where $\mathcal{H}$ is a *specific structured class of distributions* such as mixtures of Gaussians [KMV12, DK14, SOAJ14, KSS18, ABM18, ABH+20], histograms [Pea95, CDSS14, CDKL22], and polynomials [ADLS17]. The abstract hypothesis selection we study here is commonly used as a subroutine in solving these problems

---

[1]It is worth noting that we do not have any pre-processing step in this paper.

(usually in conjunction with some sort of a cover method). For a survey of results, see [Dia16].

Lastly, there is another field of study that tackles a question similar to ours: the problem of sorting items with noisy comparisons. Defining $val(H) := \|H - P\|_{TV}$, we can view hypothesis selection as the task of approximately minimizing $val(H_i)$ over $i \in [n]$ given a noisy comparator for these values. Multiple studies have been conducted in various noise models such as [BM09, AFHN15, AFJ+18, WGW22]. In the work, the outcomes of the comparisons are randomized, e.g., flipping the true outcome of a comparison with a fix probability, or randomly declaring $x > y$ with probability $x/(x + y)$. With the exception of [AFJ+18] that we have discussed above, these probabilistic noise models do not capture the geometric structure presenting in our problem. Our comparisons are consistent with the probability masses in the subsets of the domain imposing some deterministic constraints on the comparisons. For example, an excellent hypothesis does not lose to $H_i$ if $\|H_i - P\|_{TV} > 3 \max(\Gamma, \mathsf{OPT})$.

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

$$\leq \exp\Big(-\frac{k \cdot p_0}{8}\Big) + \frac{\delta}{2} \leq \exp\Big(-\frac{1}{2\,\delta}\Big) + \frac{\delta}{2} \leq \frac{3\,\delta}{4},$$

(By the Hoeffding inequality)

where the last inequality holds due to the fact that $e^{-1/(2\,\delta)} < \delta/4$ for $\delta \leq 0.1$. Hence, the proof is complete. □

## 3.3  Memory-data tradeoffs of random ladder tournament

As we have discussed, one of the advantages of Algorithm 2 is its flexibility in the usage of memory and data. In the following, we describe a tradeoff that we can obtain for this algorithm using the sample summaries presented in Section 2.2.2. At a high-level, the following is how we achieve the tradeoff: suppose we have $b$ bits of memory. We choose the largest integer $t$ so that we can store the sample summary needed to compare $t$ hypotheses. Then, we run the random ladder tournament while we draw new samples and refresh the sample summary at every $t$ step. Our two described sample summaries Scheff'e counts and the sorted list lead to the following memory-data tradeoffs. Although these tradeoffs are not as tight as our main result (by factors of $\log n$ and $1/\epsilon$), they have better accuracy guarantees ($\alpha = 3$ instead of $\alpha = 9$ in our main result).

**Lemma 3.2.** *Suppose we have $n$ hypotheses in $\mathcal{H}$. For every $p_0 \geq 1/n$, $\epsilon$, and an integer $t$ between two and $k = \Theta(1/p_0)$, one can implement Algorithm 2 in such away that it uses:*

$$s = O\Big(\frac{1}{t\,p_0} \cdot \frac{\log p_0^{-1}}{\epsilon^2}\Big) \ \text{ samples, and } \quad b = O\Big(t^2 \log\Big(\frac{\log p_0^{-1}}{\epsilon}\Big) + t \log n\Big) \ \

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

□