# OpenReview forum: "Hypothesis Selection with Memory Constraints"
_NeurIPS.cc/2023/Conference — NeurIPS 2023 poster_

### Official Review · Reviewer_8owE · 2023-06-13

**Soundness:** 4 excellent
**Presentation:** 3 good
**Contribution:** 3 good
**Rating:** 7
**Confidence:** 4

**Summary:**

The paper studies hypothesis selection in a streaming setting, with samples arriving online. The main result gives a near-optimal memory-sample trade-off for hypothesis selection.  Along the way, the paper invents several new techniques to avoid expensive memory usage of prior work.

**Strengths:**

This is one of the most classic questions in nonparametric statistics. Prior work [MS08, DK14] have tackled the run-time problem. I believe that its memory efficiency is also an important aspect. This paper offers exactly such a study.

At a technical level, the paper nearly resolves the question, up to some minor logarithmic factors. The high-level approach is simple and may be practical. The techniques, as far as I know, are novel and could potentially be used by later works. I have not checked the proof details in the appendix, but based on the main body, the main arguments are sound.

The paper is generally well-written.


**Weaknesses:**

I do not find any major weakness of this paper.

One thing I should point out is that the imperfect comparison issue (discussed in section 2.2.1) has been studied in the literature of noisy sorting. In fact, I believe the main result from the random ladder tournament, *at least* in the offline setting (section 3.1), can be derived simply in a blackbox way from Theorem 3.8 of Sorting and Selection with Imprecise Comparisons by Miklós Ajtai, Vitaly Feldman, Avinatan Hassidim and Jelani Nelson (TALG 2015) (https://dl.acm.org/doi/10.1145/2701427). In particular, their $\delta$ is your $3\Gamma$, the error gap between good and bad hypotheses in the comparison procedure (Algorithm 1). Their number of items there is your number of hypotheses, both denoted by $n$. So up to another factor of $k=3$, applying their randomized max-finding algorithm would yield a hypothesis that’s $9\Gamma$ close to the optimal.  (Another related work is https://arxiv.org/abs/1606.02786 where an improved algorithm is given and an application to hypothesis selection is explicitly derived. I have not checked the details, though.)

I would ask the author(s) to confirm this and see if an alternative argument using the paper above can be made and whether it’s interesting.

I should note that this, if true, does not trivialize the results in this paper. The algorithm of Ajtai et al cannot be implemented in the memory-constrained setting, below a memory size of $n^{1/3}$. Also in the offline setting, the resulting constant is not optimal for the hypothesis selection problem.




Minor comments
---
Line 62–67: list these two query access assumptions as \begin{itemize}

Line 65: “ the probabilities of the Scheffé sets”  — I think it’s more clear to refer to this as “the probability masses of the Scheffe sets”.

Line 117: “we sample a uniformly random new hypothesis from H” — Please clarify: is this sampling with replacement, if I view the meta-distribution as uniform over the n input hypotheses? That is, a hypothesis can be selected multiple times against the same single hypothesis in memory.

Line 202: Regarding [MS08], mention that the simple linear-time selection from Appendix A doesn’t do such expensive preprocessing.

Line 265: Can you clarify in which scenario or parameter regime would you use Scheffé counts? (I didn’t read the appendix.) Note that the memory usage of this technique is quadratic in $n$. Hence, are you tracking these counts on a small subset of hypotheses?

**Questions:**

To avoid the assumption of having access to the probability of Scheffe sets, have you considered approaches that don’t require Scheffe sets at all? I understand that the minimum distance estimate [DL01, MS08] is used in the all-go-against-all tournament step. Can ideas like that be used throughout, so we don’t need query Scheffe sets?


**Limitations:**

The author(s) have discussed some future directions, though I think a conclusion section summarizing these would be great.

---

> ### Author Rebuttal · Authors · 2023-08-09
>
> > the main result from the random ladder tournament, ... can be derived simply ... from ...(https://dl.acm.org/doi/10.1145/2701427).
>
> Thank you for the reference. We will be sure to add it to our discussion of previous work. That said, we did not see how the main result can be derived in a blackbox way from Theorem 3.8.
>
> Defining $val(H) := \|H - P\|_{tv}$, we can view hypothesis selection as the task of approximately minimizing $val(H_i)$ over $i \in [n]$ given a noisy comparator for these values. To apply their Theorem 3.8 to this task, we need to identify the distance parameter $\delta$ for which our COMPARE subroutine (our Theorem 1) is always guaranteed to produce the correct comparison between two hypotheses whose values are at least $\delta$ far apart. For every pair of hypotheses $H_i, H_j$, our COMPARE subroutine correctly reports that $val(H_i) \ge val(H_j)$ as long as $val(H_i) \ge 3 \cdot val(H_j)$, and this is tight.
>
> Thus, the distance parameter we can achieve is lower bounded by $\delta \ge \min\{val(H_i), val(H_j)\}$. Taking the maximum over all pairs $i, j$, we see that the best (smallest) $\delta$ we can achieve is the second largest value of $val(H_i) = \|H_i - P\|_{tv}$, taken over all $i \in [n]$. This can be arbitrarily large as a multiple of $OPT = \min_{i \in [n]} \|H_i - P\|_{tv}$, and so approximating the minimum value to within $O(\delta)$ may not bring us within $O(OPT)$.
>
> Of course, would be happy to be corrected on this point if the reviewer has a different argument in mind.
>
> -----------------------------
>
> > Another related work is https://arxiv.org/abs/1606.02786 where an improved algorithm is given ...
>
> This paper (henceforth "AFJOS16") is absolutely relevant, and we thank the reviewer for pointing it out.
>
> AFJOS16 contains an algorithm for solving hypothesis selection (density estimation) making linearly many comparisons and running in $O(n\log n)$ time, that achieves accuracy ratio $\alpha = 9$.
>
> As far as we understand, our algorithms improve on AFJOS16 in two respects.
> 1. It is not straightforward how to implement the AFJOS16 algorithms with $o(n)$ bits of memory. In particular in their "Q-Select" algorithm, one maintains a set of hypotheses of size $\Omega(n)$.
> 2. The accuracy parameter $\alpha$ we obtain is better (smaller) than that of AFJOS16. Note that AFJOS16 solves the density estimation problem without promise (no knowledge of $\Gamma$), so to draw a fair comparison, we make this statement about the "no promise" version of our algorithm achieving $\alpha = 5$ (which is less than the factor of 9 they achieve). See Section A, Theorem A.2 of our submission.
>
> We will make sure to cite and discuss this paper in future versions of the paper.
>
> -----------
>
> > Line 117: “we sample a uniformly random new hypothesis from H” — Please clarify: is this sampling with replacement,...
>
> Good question. We draw the hypotheses from a meta distribution in an i.i.d. manner. If the meta distribution is a uniform distribution over $\mathcal{H}$, then we sample uniformly from $\mathcal{H}$ with replacement, implying that hypotheses can be selected several times. These repetitions  do not affect our analysis as long as we ensure that $H^*$ shows up with probability at least $p_0$ at every step.
>
> We use the generality of the meta-distribution in our main result, where we apply the ladder to the (not necessarily uniform) distribution on hypotheses that comes from the output of a particular subroutine.
>
> A second reason is that the meta-distribution version of the ladder allows to amplify the success probabiltiy of the algorithm by running for more steps without changing the approximation guarantee. (Naive ways of amplifying success probability require another layer of comparisons, which changes the approximation constant.)
>
> We will edit our text to make sure these points are clear.
>
> -------------
>
> >Line 265: Can you clarify in which scenario or parameter regime would you use Scheffé counts? ...
>
>
> Your observation on quadratic memory use of Scheffé counts is right. For algorithm with $b$ bits, we focus on $t \approx \tilde{O}(\sqrt{b})$ hypotheses. $t$ is chosen so $O(t^2)$ Scheffé counts fit in $\Theta(b)$ bits for any comparison among $t$ hypotheses.
>
> We have used the Scheffé counts for two results:
> 1) A basic tradeoff presented in Lemma C.1
> 2) Main result presented in Section E.4.
>
> We discuss each of these in a bit more detail in case it is helpful:
> 1. We start by off the sketch for our basic tradeoff: We store the Scheffé counts for $t$ random hypotheses. In this way, we can easily compare these $t$ hypotheses, so we can run the random ladder tournament for $t$ steps. We repeat this process again. We take the winner of the last step  and add $t-1$ random hypotheses. Draw fresh sample and obtain the Scheffé counts of these $t$ hypotheses, and move forward with another $t-1$ steps of the random ladder tournament. This approach leads to a tradeoff described in Lemma C.1. This tradeoff is off by a factor of $t \approx \sqrt{b}$ due to the inefficincy of the Scheffe counts. However, this tradeoff has a better dependency on $\epsilon$ (and in some way on $\log n$) than what one gets from  Lemma 3.1: in Lemma C.1 we have $b \approx O(t^2)$ instead of $b \approx O(t \log n/\epsilon^2)$ in Lemma 3.1.
>
> 2. We exploit this advantage of Scheffé count later in the proof of our main result. The reason that we can tolerate this extra factor of $\sqrt{b}$ is that in the final algorithm the random ladder is used on only $k$ filtered hypotheses where $k
> \ll n$ (roughly, $k \approx \sqrt{n})$. Hence, we have some slack to tolerate $\sqrt{b}$. On the other hand, if we use the other simple tradeoff (Lemma 3.1) that is based on sorted lists, we get a factor of $1/\epsilon^4$ that we cannot improve. Therefore, while the simple tradeoff of Lemma C.1 is worse than Lemma 3.1, this is the tradeoff we used in our main theorem. See Section E.4 in the supplementary material for more details.

---

> > ### Comment · Reviewer_8owE · 2023-08-10
> >
> > Thank you for the response! I maintain my rating and recommend accept.
> >
> > Could you fix the LaTex in your rebuttal?

---

> > > ### Author Response · Authors · 2023-08-10
> > >
> > > Thank you for your prompt response. We apologize for the formatting issues. Since the deadline has passed, we cannot edit our text. However, we have added that part of our response below:
> > >
> > > > the main result from the random ladder tournament, ... can be derived simply ... from ...(https://dl.acm.org/doi/10.1145/2701427).
> > >
> > > Thank you for the reference. We will be sure to add it to our discussion of previous work. That said, we did not see how the main result can be derived in a blackbox way from Theorem 3.8.
> > >
> > > Defining $val(H) := \|H - P\|_{tv}$, we can view hypothesis selection as the task of approximately minimizing $val(H_i)$ over $i \in [n]$ given a noisy comparator for these values. To apply their Theorem 3.8 to this task, we need to identify the distance parameter $\delta$ for which our COMPARE subroutine (our Theorem 1) is always guaranteed to produce the correct comparison between two hypotheses whose values are at least $\delta$ far apart. For every pair of hypotheses $H_i, H_j$, our COMPARE subroutine correctly reports that $val(H_i) \ge val(H_j)$ as long as $val(H_i) \ge 3 \cdot val(H_j)$, and this is tight.
> > >
> > > Thus, the distance parameter we can achieve is lower bounded by $\delta \ge 2 \cdot \min\left(val(H_i) , val(H_j)\right)$. Taking the maximum over all pairs $i, j$, we see that the best (smallest) $\delta$ we can achieve is the second largest value of $2\cdot val(H_i) = 2\cdot \|H_i - P\|\_{tv}$, taken over all $i \in [n]$. This can be arbitrarily large as a multiple of $OPT = \min_{i \in [n]} \|H_i - P\|_{tv}$, and so approximating the minimum value to within $O(\delta)$ may not bring us within $O(OPT)$.
> > >
> > > Of course, would be happy to be corrected on this point if the reviewer has a different argument in mind.

---

> > > > ### Comment · Reviewer_8owE · 2023-08-12
> > > >
> > > > Thank you for the clarifications! I have no more concerns and would recommend accept.

---

### Official Review · Reviewer_Awx8 · 2023-07-02

**Soundness:** 3 good
**Presentation:** 3 good
**Contribution:** 3 good
**Rating:** 6
**Confidence:** 2

**Summary:**

The paper studies the problem of hypothesis selection under a memory constraint. Here one is given $n$ distributions $H_1, \dots, H_n$ with access to an oracle that can output 1) $H_i(H_j > H_k)$ for any $i,j,k$ and 2) $1(H_i(x) > H_j(x))$ for any $i,j$ and any point $x$ in the underlying space. Given a stream of observations $x_1,x_2...$ from some unknown $P$ the task is to select an estimator $\hat{H} \in (H_i)_{i=1}^n$ that is as close as possible to $P$ in TV distance up to a multiplicative and additive constant $\alpha,\epsilon$ respectively. Moreover, one wishes to do so using at most $b$ bits of memory at any point during the algorithm.

The author's main result Theorem 1 shows that $s$ samples from $P$ are sufficient to output a suitable estimator with constant error probability, provided $bs \gtrsim \tilde{\mathcal \Omega}(n\log(n)/\epsilon^2)$ and note that $bs \gtrsim n$ is necessary by previous work. They achieve this result by introducing multiple technical ideas, the main one being the 'random ladder tournament'.

**Strengths:**

- The paper is mostly clearly written
- The paper appears technically sound
- The paper introduces simple but novel technical ideas that not only near-optimally tackle their proposed problem, but also recover other known results.

**Weaknesses:**

- My main concern is that due to the technical nature of the paper the short format of neurips is simply not enough for a meaningful presentation of the results. The motivating problem is under a memory constraint, yet just as one would learn how one can trade off memory for samples in the random ladder tournament (Lemma 3.1) the paper ends. More generally, from the main text, I don't feel like I have learnt how the algorithm actually works. In sections 1.2, 2 and 3 we are given glimpses of the key technical ideas necessary but it doesn't feel cohesive.


Typos:
Page 4 "Results in a better..."

**Questions:**

I would suggest to shorten (or remove) the proof of theorem 4 and to replace it with a (sketch) proof of the memory-sample trade-off.

**Limitations:**

The authors address limitations.

---

> ### Author Rebuttal · Authors · 2023-08-09
>
> > due to the technical nature of the paper the short format of neurips is simply not enough for a meaningful presentation of the results
>
> We will endeavor to present at least some of the main ideas concisely in any future short versions, at NeurIPS or elsewhere. Like many NeurIPS submissions with detailed technical components, we present full descriptions of our algorithms and their analyses in the supplementary materials.
>
> We intended Section 2.1 to explain our key ideas and the general flow of the paper. We selectively chose the random ladder tournament to be the main technical tool we present in addition to one of our basic tradeoffs presented in Lemma 3.1. We will add a more detailed discussion regarding these results in the main body of the paper (see also the response to Reviewer 5dQu).
>
> Regardless of the outcome of the NeurIPS reviewing process, we will upload the full version of our work to arXiv.

---

### Official Review · Reviewer_5jdF · 2023-07-05

**Soundness:** 4 excellent
**Presentation:** 4 excellent
**Contribution:** 3 good
**Rating:** 6
**Confidence:** 3

**Summary:**

The authors study the problem of hypothesis testing for pdfs in a streaming model. The problem is to find a hypothesis H* (corresponding to a pdf) from a family {H1,...,Hn} that is closest to an unknown pdf P. The input is a stream of points drawn i.i.d. from P. At any time step the algorithm can ask for a new point or query a scheffe set of two distributions Hi and Hj in {H1,..,Hn}.

The main theorem in the paper shows that the algorithm can "learn" P properly by using O(n\log n/b\cdot\frac{\log(1/\eps)}{\eps^2}) queries, where b is the memory used. This is close to optimal - within log n factor.

Note:
- proper learning a pdf means: |P-H*|_TV\le\alpha\cdot OPT({H1,...,Hn}, P)+\eps.
- scheffe set of two distributions H1, H2 is {x : H1(x)<H2(x)}

**Strengths:**

- Clean presentation and summary at the beginning of the paper.
- Almost tight bounds (upto log factors)

**Weaknesses:**

- Perhaps the model can be motivated better? Why the restriction to scheffe sets?
- Worth comparing the algorithm to vast statistical literature on Neyman-Pearson tests using log-likelihood computation.
- Need to compare and cite some of the vast literature on orienteering tournaments in directed graphs - I suspect some of the underlying results may already be known.

**Questions:**

Why not design an online Neyman-Pearson test - compute approximate log likelihoods based on the query results? The log-likelihood can then be used to upper bound the TV distance.

**Limitations:**

NA. The paper is theoretical in nature and is fairly upfront about it.

---

> ### Author Rebuttal · Authors · 2023-08-09
>
>
> > Why the restriction to scheffe sets?
>
> Scheffé set queries provide a generic computational model for expressing algorithms that apply to many families of distributions; this avoids assumptions on particular data formats or functional forms for the distributions. Scheffé queries are also sufficient for implementing existing algorithms for hypothesis selection. The fact that our algorithms only need to access the distribution family $\mathcal{H}$ using Scheffé set queries makes our positive results stronger.
>
> We note that the nearly-matching lower bounds, based on communication-complexity arguments, apply to all algorithms -- in particular, including those not based on Scheffé sets.
>
> > Need to compare and cite some of the vast literature on  orienteering tournaments in directed graphs
>
> We appreciate the suggestion. Unfortunately, we had trouble identifying which technical ideas would be relevant to our submission. We searched for papers on orienteering, but only found work on a class of NP-complete optimization problems that are variants on the Traveling Salesman problem. (A representative example is "The Directed Orienteering Problem" by Nagarajan and Ravi in *Algorithmica* 2011.) Perhaps the reviewer could point us to a starting point for the literature they had in mind?
>
> > vast statistical literature on Neyman-Pearson tests
>
> We will add some further discussion of this literature. However, note that maximum likelihood approaches do not generally work well for hypothesis selection in total variation ($\ell_1$) distance, even when we just want to select among two distributions. This is discussed, for example, in the textbook of Devroye and Lugosi, Section 6.4. We reproduce their counterexample here.
>
> Suppose $H_1$ is a uniform distribution over $[-1, 1]$, and $H_2$ is a uniform distribution over $[\delta, 1+\delta]$ for some parameter $\delta \in [0,0.5)$ that we determine later. Let $P$ be the uniform distribution over $[0,1]$. Then
> $$0.5 = \|H_1 - P\|_{tv} > \|H_2 - P\|_{tv} = \delta\,.$$
> The correct output for the selection problem is  $H_2$ once $\delta$ is sufficiently small (say, sub-constant). However, the MLE will select $H_1$ if any  of the data samples fall in $[0,\delta]$. For $\delta \gg 1/s$, where $s$ is the number of samples, we will see a sample in $[0,\delta]$ with high probability. There is thus a fairly wide range of $\delta$ for which  the maximum likelihood estimate will be incorrect.

---

### Official Review · Reviewer_5dQu · 2023-07-11

**Soundness:** 3 good
**Presentation:** 2 fair
**Contribution:** 3 good
**Rating:** 6
**Confidence:** 3

**Summary:**

This paper studies the problem of agnostic distribution learning where i.i.d. samples are generated from an unknown distribution $X$. The goal is to find the best distribution from a given set of finite distributions $\\{H_1,\cdots,H_n\\}$ that is closest to $X$ under total variation distance. The authors specifically study the sample complexity, under which the memory that is needed to store the information of the samples is bounded. The main contribution is a trade-off between the number of bits of the memory and samples needed, which is tight up to a $\log n$ factor in the minimax sense.

The main proof technique is based on the so called "random ladder tournament". This differs from conventional approach that first estimating the probability of Scheffé sets of all pairs in $\\{H_1,\cdots,H_n\\}$ and then selecting a distribution closest to the estimates. In contrast, the authors propose an ingenious streaming algorithm that tracks the "best" distribution in an online fashion, which is suited for achieving their desired trade-off between memory usage and sample complexity.

**Strengths:**

The paper's primary strength lies in its novel "random ladder tournament" algorithm. This algorithm effectively achieves a memory-efficient sample complexity for agnostic distribution learning. The approach incorporates some interesting technical ideas that may be of independent  interest. As far as I'm aware, this particular scenario has not being studied in the literature.

**Weaknesses:**

My main concern regarding this paper is its overall significance to the machine learning community. The paper has a strong TCS flavor, and the introduced model appears somewhat contrived. Specifically, the derived trade-offs, though mathematically intriguing, do not offer any surprising insights. Thus, its practical implications and utility within a broader machine learning context may be limited.

Additionally, the paper's presentation could certainly benefit from some improvements. The authors dedicate a significant portion of the introduction to outlining their techniques, which, unfortunately, are quite challenging to grasp without delving into the substantive technical details found mainly in the appendix.

I would recommend eliminating most of this preliminary discussion and focusing primarily on the "random ladder tournament" argument, as outlined in Section 3. Providing a more detailed and accessible explanation of this central argument could make the paper more digestible for readers. Following this, a more straightforward discussion of the techniques used to improve the logarithmic terms would be better placed and easier to understand. This reordering should contribute to a more coherent and engaging narrative throughout the paper.

**Questions:**

See comments above.

**Limitations:**

No issue with negative societal impact.

---

> ### Author Rebuttal · Authors · 2023-08-09
>
> > overall significance to the machine learning community
>
> Hypothesis selection is a fundamental problem in statistical learning theory. The formulation we study here abstracts and generalizes many specific distribution selection/estimation tasks (some of which are discussed in the book of Devroye and Lugosi, "Combinatorial Methods in Density Estimation").
>
> Meanwhile, when one takes a computational perspective, memory and communication are important resources to control in practical implementations, and their study gives rise to a variety of theoretical questions.
>
> Our goal in formulating this problem was to study memory/sample tradeoffs in a general setting, and we hope our results serve as a launching point for studying these resources systematically.
>
> Note that general hypothesis selection algorithms (i.e., ones that do not use information specific to the distributions under consideration) play an important role in several model-specific learning learning problems. This includes the Gaussian mixture-learning algorithms of Daskalakis and Kamath. Papers on this topic regularly appear at NeurIPS. Examples include:
>
> [Private hypothesis selection](https://papers.nips.cc/paper_files/paper/2019/hash/9778d5d219c5080b9a6a17bef029331c-Abstract.html). M Bun, G Kamath, T Steinke, S Z  Wu. *NeurIPS 2019*
> [Nearly tight sample complexity bounds for learning mixtures of gaussians via sample compression schemes](https://proceedings.neurips.cc/paper_files/paper/2018/hash/70ece1e1e0931919438fcfc6bd5f199c-Abstract.html). Ashtiani, S. Ben-David, N. Harvey, C. Liaw, A. Mehrabian, and Y. Plan. *NeurIPS 2018*
> [Near-Optimal-Sample Estimators for Spherical Gaussian Mixtures](https://proceedings.neurips.cc/paper/2014/hash/c0f168ce8900fa56e57789e2a2f2c9d0-Abstract.html). A T Suresh, A Orlitsky, J Acharya, A Jafarpour. *NIPS 2014*
> [Near-optimal density estimation in near-linear time using variable-width histograms](https://papers.nips.cc/paper_files/paper/2014/hash/287e03db1d99e0ec2edb90d079e142f3-Abstract.html). S Chan, I Diakonikolas, R Servedio, X Sun. *NIPS 2014*
>
>
>
> > I would recommend [...] focusing primarily on the "random ladder tournament"
>
> That's a good suggestion. It agrees with some of the comments from other reviewers. We will try to implement it in future short versions of the paper (whatever the outcome at NeurIPS).

---

> > ### Comment · Reviewer_5dQu · 2023-08-11
> >
> > I thank the authors for addressing my concerns. After reviewing the rebuttals and feedback from other reviewers, I am convinced that the model studied holds inherent theoretical value and offers potential practical utility. I trust the authors to address the presentation issues to the best of their ability. I have also decided to adjust my rating upward.

---

### Author Rebuttal · Authors · 2023-08-09

We thank the reviewers for their time and thoughtful comments. We will incorporate all the editorial comments.
Please find the individual responses to the reviewers below.

---

### Decision · Program_Chairs · 2023-09-21

**Decision:**

Accept (poster)

**Comment:**

The authors study the classic problem of hypothesis selection but under memory constraints. They define a certain query model where the leaner can compare the pdf values of the distributions at the given data point. The authors then present an algorithm that achieves (near) optimal tradeoff between memory and sample complexity in a certain sense: the number of bits (memory) times the number of samples used is O(n.log n), matching a known lower bound.

The reviewers were quite positive about the paper, and found the contributions significant. There was some doubt about the relevance of the algorithmic (i.e., memory efficiency) contributions to the NeurIPS community. There were also concerns about the quality of the presentation and the reviewers thought the setting (the query model as well as the tradeoff measure) could be better justified. The authors are therefore encouraged to elaborate more on the model, and perhaps discuss the distributed learning under information constraints angle in more details.

I recommend acceptance.